

# Incorporating modelled subglacial hydrology into inversions for basal drag

Conrad P. Koziol[1,2] and Neil Arnold[1]

[1]Scott Polar Research Institute, Cambridge U.K.
[2]now at the University of Edinburgh

*Correspondence to:* Conrad P. Koziol (ckoziol@gmail.com)

**Abstract.** A key challenge in modelling coupled ice flow - subglacial hydrology is initializing the state and parameters of the system. We address this problem by presenting a workflow for initializing these values at the start of a summer melt season. The workflow depends on running a subglacial hydrology model for the winter season, when the system is not forced by meltwater inputs, and ice velocities can be assumed constant. Key parameters of the winter run of the subglacial hydrology

model are determined from an initial inversion for basal drag using a linear sliding law. The state of the subglacial hydrology model at the end of winter is incorporated into an inversion of basal drag using a non-linear sliding law which is a function of water pressure. We demonstrate this procedure in the Russell Glacier Area, and compare the output of the linear sliding law with two non-linear sliding laws. Additionally, we compare the winter output of a recent subglacial hydrology model to radar observations, and find that the modelled state of the subglacial hydrology system at the end of winter is in line with summer

rather than winter observations.

## 1   Introduction

Subglacial hydrology is an important control on ice velocities at the margin of the Greenland Ice Sheet. Observed seasonal acceleration of ice flow (Joughin et al., 2008; van de Wal et al., 2008; Zwally et al., 2002) is driven by the evolution of subglacial system between distributed and channelized states in response to meltwater input (Bartholomew et al., 2010; Chandler et al.,

2013; Cowton et al., 2013; Schoof, 2010). However, the impact of melt season intensity on seasonal and annual velocities, and how it may change in the future, is not fully understood. Observational studies of land-terminating sectors of the Greenland ice sheet reveal a complex set of possible interactions. Increased surface melt may result faster flow early in the melt season, offset by a stronger late summer deceleration (Sundal et al., 2011). Increased runoff may also lead to more extensive drainage of the ice sheet base, reducing annual velocities due to slower winter flow (Sole et al., 2013). Long term observations in the

ablation zone show surface melt and ice velocities are anti-correlated over decadal timescales (Stevens et al., 2016; Tedstone et al., 2015; van de Wal et al., 2015). The possible impact of surface melt on ice velocities at higher elevations is less well understood, as is the impact in marine-terminating sectors.

Recent subglacial hydrology models have progressed to simultaneously incorporating both distributed and efficient systems, explicitly treating the interaction between the two (de Fleurian et al., 2016; Hewitt, 2013; Pimentel et al., 2010; Schoof,

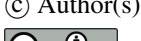



2010; Werder et al., 2013). These models have shown success in recreating the broad pattern of subglacial development in the summer melt-season inferred from GPS measurements (Bartholomew et al., 2011; van de Wal et al., 2015) and dye-tracing experiments (Chandler et al., 2013; Cowton et al., 2013). The development of the subglacial hydrological system has been shown to depend on feedbacks from ice velocities (Hoffman and Price, 2014). However, applications of recent hydrology

models coupled with ice-flow models have been limited to idealized domains (Hewitt, 2013; Hoffman and Price, 2014; Hoffman et al., 2016; Pimentel and Flowers, 2010).

Initializing model parameters and state is necessary for applying a linked hydrology/ice dynamics model to the Greenland Ice Sheet. In contrast to the availability of measurements at the surface of ice sheets however, the conditions at the ice-bed interface are poorly constrained. Some key challenges for modelling are: the form of the sliding law which relates water pressures to

basal drag; the values of the parameters in that relationship; the values of water pressures at the ice-bed interface.

Inverse methods are an approach which can be used to constrain unknown variables or parameters in an ice sheet model. Inversions optimize the value of an unknown to minimize the discrepancy between model output and observed data. Since basal drag and the parameters of the sliding law are one of the least constrained inputs to ice sheet models, a common application of inversions in glaciology is to determine the field of basal drag which best reproduces observed surface velocities. A variety

of inversion methodologies have been applied in glaciology. These include iterative methods (Arthern et al., 2015), automatic differentiation (Goldberg and Heimbach, 2013; Heimbach and Bugnion, 2009; Martin and Monnier, 2014), and lagrangian multiplier methods based on control theory (MacAyeal, 1993; Morlighem et al., 2013).

In this study we develop an ice sheet model and inversion code which we apply to the Russell Glacier region of Western Greenland in order to invert for basal drag at the end of winter. The ice sheet model uses the hybrid formulation of Arthern

et al. (2015) and Goldberg (2011) and is numerically similar to Arthern et al. (2015). The inversion procedure is based on automatic differentiation (Goldberg and Heimbach, 2013). Our application is novel in that we constrain the inversions using the output of a current subglacial hydrology model (Hewitt, 2013), and we investigate the impact of three different sliding laws (linear, generalized Weertman and Schoof-type (Hewitt, 2013)) on the patterns of basal drag predicted by the model. These developments form an important preliminary step towards a fully coupled model for ice dynamics and glacier hydrology which

can be validated using current observational ice velocity data, and subsequently used for prognostic studies of possible future ice sheet responses to increased surface melt.

## 2    Methods

### 2.1    Hybrid Ice Sheet Model

#### 2.1.1    Model Formulation

The ice sheet model implemented is based on the hybrid formulation described in Goldberg (2011) and Arthern et al. (2015), and uses the numerical implementation of Arthern et al. (2015).





Following Goldberg (2011) and Arthern et al. (2015), the conservation of momentum equations for depth-averaged velocities are:

$$\partial_x(4h\bar{\eta}\partial_x\bar{u} + 2h\bar{\eta}\partial_y\bar{v}) + \partial_y(h\bar{\eta}\partial_x\bar{v} + h\bar{\eta}\partial_y\bar{u}) - \tau_{bx} = \rho_i g h \partial_x s \tag{1}$$

$$\partial_y(4h\bar{\eta}\partial_y\bar{v} + 2h\bar{\eta}\partial_x\bar{u}) + \partial_x(h\bar{\eta}\partial_y\bar{u} + h\bar{\eta}\partial_x\bar{v}) - \tau_{by} = \rho_i g h \partial_y s \tag{2}$$

where $u(x,y,z)$ and $v(x,y,z)$ are velocities in the x and y directions, $\eta(x,y,z)$ is dynamic viscosity, $h(x,y)$ is ice thickness, $s(x,y)$ is surface elevation, $\tau_{bx(x,y)}$ and $\tau_{by}(x,y)$ are basal drag in the x and y directions, g is the magnitude of gravitational acceleration, and $\rho_i$ is the density of ice. The overbar ($\bar{x}$) denotes the depth averaged value of a variable, so that $\bar{u}(x,y)$ and $\bar{v}(x,y)$ are depth averaged velocities and $\bar{\eta}(x,y)$ is depth averaged viscosity.

Basal drag is defined by the sliding law. Three different sliding laws are implemented in the ice sheet model:

$$\boldsymbol{\tau_b} = \beta^2 \boldsymbol{u_b} \tag{3}$$

$$\boldsymbol{\tau_b} = \mu_a N_+^p U_b{}^q \frac{\boldsymbol{u_b}}{U_b} \tag{4}$$

$$\boldsymbol{\tau_b} = \mu_b N_+ \left(\frac{U_b}{U_b + \lambda_b A_b N_+^n}\right)^{\frac{1}{n}} \frac{\boldsymbol{u_b}}{U_b} \tag{5}$$

where $\boldsymbol{\tau_b} = (\tau_{bx}(x,y), \tau_{by}(x,y))$ is the basal drag, $\boldsymbol{u_b} = (u_b, v_b) = (u(x,y,b), v(x,y,b))$ is the basal velocity, $U_b$ is the sliding speed ($|\boldsymbol{u_b}|$), $N(x,y) = \rho_i g h - p_w$ is the effective pressure at the ice sheet bed, $p_w$ is water pressure, $\beta(x,y)$ is a basal drag coefficient, $\mu_a(x,y)$ is a drag coefficient, p and q are positive exponents, $\mu_b(x,y)$ is a limiting roughness slope, $\lambda_b$ is the characteristic bed roughness length, and $A_b$ and n are coefficients in Glen's flow law (Hewitt, 2013). $A_b$ is the ice creep parameter set to an appropriate value for basal ice. Following Hewitt (2013), negative effective pressures are eliminated by setting $N_+ = max(N, 0)$, and regularized with a small regularization constant.

The linear sliding law (Equation 3) represents all ice-bed interactions by a single friction coefficient $\beta$. The second and third equations are a generalized Weertman sliding law and a Schoof sliding law respectively (Hewitt, 2013). These attempt to explicitly represent more complex interactions at the ice-bed-interface, in particular, the impact of basal water pressure. Equation 4 is a power law commonly used in glaciology to describe basal rheology (e.g. Bueler and Brown, 2009; MacAyeal, 1989; Hewitt, 2013), although typically with no dependence on effective pressure ($p = 0$). At high effective pressures the Schoof sliding law has a similar form ($\boldsymbol{\tau_b} \approx \mu_b(\lambda_b A_b)^{-1} U_b^{\frac{1}{n}}$), but transitions to a Coulomb description at low effective pressures ($\boldsymbol{\tau_b} \approx \mu_b N$).





It is useful to represent the sliding laws in a common form:

$$\boldsymbol{\tau_b} = C\boldsymbol{u_b} \tag{6}$$

where C is a function multiplying basal velocities. The form and parameters of C depend on the sliding law.

The boundary conditions at the terminating margin of the ice sheet are:

$$2\bar{\eta}h(2\partial_x\bar{u} + \partial_y\bar{v})\hat{n}_x + \bar{\eta}h(\partial_y\bar{u} + \partial_x\bar{v})\hat{n}_y = \frac{g}{2}(\rho_i h^2 - \rho_w d^2)\hat{n}_x \tag{7}$$

$$2\bar{\eta}h(2\partial_y\bar{v} + \partial_x\bar{u})\hat{n}_y + \bar{\eta}h(\partial_y\bar{u} + \partial_x\bar{v})\hat{n}_x = \frac{g}{2}(\rho_i h^2 - \rho_w d^2)\hat{n}_y \tag{8}$$

where $\rho_w$ is the density of water, d is the ice draft (zero at land terminating portions of the margin), and $\hat{n}_x$ and $\hat{n}_y$ are the components of the unit vector normal to the terminating margin (Goldberg, 2011; Arthern et al., 2015).

Two further boundary conditions are used in the ice sheet model: a no-penetration condition at the margin of nunatuks, and a dirichlet boundary condition at the lateral margins of the ice sheet domain which are not the termination edge.

The equation for viscosity is:

$$\eta = \frac{1}{2}A^{\frac{-1}{n}}\left((\partial_x u)^2 + (\partial_y v)^2 + (\partial_x v)(\partial_y u) + (\partial_x v + \partial_y u)^2 + \frac{1}{4}(\partial_z u)^2 + \frac{1}{4}(\partial_z v)^2 + \epsilon_0\right)^{\frac{1-n}{2n}} \tag{9}$$

where $\epsilon_0$ is a regularization term. Vertical shearing in the hybrid formulation is approximated by:

$$\partial_z u \approx \partial_z u + \partial_x w = \frac{\sigma_{xz}}{\eta}, \quad \partial_z v \approx \partial_z v + \partial_y w = \frac{\sigma_{yz}}{\eta} \tag{10}$$

As in Goldberg (2011) and Arthern et al. (2015), a linear relationship between vertical shear stresses and depth is assumed:

$$\sigma_{xz} = \tau_{bx}\frac{s-z}{h}, \quad \sigma_{yz} = \tau_{by}\frac{s-z}{h} \tag{11}$$

Viscosity is defined implicitly by Equation (9). With the standard choice of n=3, this is a cubic equation, and can be solved exactly. Alternatively, a previous value of viscosity can be used to calculate an updated value. This process can be iterated upon, to create a fixed point-iteration. The default procedure in the model is to do two iterations (Koziol, 2017).

The hybrid formulation of the conservation of momentum equations depend on depth integrated viscosity:

$$\bar{\eta} = \frac{1}{h}\int_s^b \eta dz \tag{12}$$





This integral, and others, are numerically integrating using the Composite Simpson's Law.

Following Arthern et al. (2015), the following integral is defined:

$$F_a = \int_s^b \frac{1}{\eta} \left( \frac{s-z}{h} \right)^a dz \tag{13}$$

This integral can be used to define expressions for surface velocity in terms of basal velocity, and basal velocity in terms of
depth averaged velocity (Arthern et al., 2015):

$$\boldsymbol{u_s} = \boldsymbol{u_b}(1 + CF_1) \tag{14}$$

$$\bar{\boldsymbol{u}} = \boldsymbol{u_b}(1 + CF_2) \tag{15}$$

where $F_1$ and $F_2$ are determined using Equation 13 .

Additionally, defining $C_{eff}$ as follows,

$$C_{eff} = \frac{C}{1 + CF_2} \tag{16}$$

leads to an expression for basal drag in terms of depth averaged velocity (Goldberg, 2011; Arthern et al., 2015):

$$\boldsymbol{\tau_b} = C_{eff} \boldsymbol{u_b} \tag{17}$$

### 2.1.2 Model Implementation

As in Arthern et al. (2015), Equation 1 and 2 can be written in the following form:

$$\mathcal{L}(\bar{\boldsymbol{u}})\bar{\boldsymbol{u}} = \boldsymbol{f} \tag{18}$$

where:

$$\mathcal{L} = \begin{bmatrix} \partial_x 4h\bar{\eta}\partial_x + \partial_y 2h\bar{\eta}\partial_y - C_{eff} & \partial_x 2h\bar{\eta}\partial_y + \partial_y h\bar{\eta}\partial_x \\ \partial_y 2h\bar{\eta}\partial_x + \partial_x h\bar{\eta}\partial_y & \partial_y 4h\bar{\eta}\partial_y + \partial_x h\bar{\eta}\partial_x - C_{eff} \end{bmatrix} \tag{19}$$

and

$$\boldsymbol{f} = \begin{bmatrix} \rho_i gh\partial_x s \\ \rho_i gh\partial_y s \end{bmatrix} \tag{20}$$




Equation 18 is a non-linear equation for depth integrated velocity. The non-linearity arises since depth integrated viscosity is a function of velocity, and in the case of a non-linear sliding law, since $C_{eff}$ is also a function of velocity. The ice sheet model solves Equation 18 on an Arakawa-C finite difference grid using a Picard iterative process.

Equation 18 is discretized following Arthern et al. (2015). The primary difference is that operators are appropriately extended to allow for periodic boundary conditions in the ISMIP-HOM experiments (Pattyn et al., 2008). Discretization of Equation 18 results in a linear system of equations, which can be written as:

$$\boldsymbol{L\bar{x}} = \boldsymbol{b} \tag{21}$$

where the matrix ($\boldsymbol{L}$) corresponds to the operator $\mathcal{L}$, while the vector $\boldsymbol{x}$ corresponds to $\bar{u}$, and the vector $\boldsymbol{b}$ corresponds to $\boldsymbol{f}$. Matlab's backslash operator is used to solve this system of equations. Alternatively, preconditioned iterative methods can be used (Arthern et al., 2015; Goldberg and Heimbach, 2013).

The Picard iteration linearizes Equation 18 by constructing $\boldsymbol{L}$ using the velocity of the previous iteration. An initial velocity guess and viscosity guess form the initial $\boldsymbol{L}$. Equation 18 is then solved for an updated velocity guess, which in turn can is used to update viscosity and $C_{eff}$. This process is repeated within a loop until the solution converges below a specified tolerance, or until a prescribed number of iterations are reached.

Evolution of surface-geometry is not included in the ice sheet model. This is appropriate since the ice-sheet model is applied on annual timescales, over which significant changes in ice sheet geometry are not expected.

The ice sheet model was tested against the ISMIP-HOM benchmark Experiments A and C (Pattyn et al., 2008), and found to compare favourably against previous models (Koziol, 2017).

## 2.2  Inversion Model

### 2.2.1  Model Formulation

This section describes the details of an inversion code developed in conjunction with the ice sheet model. The methodology is based on Goldberg and Heimbach (2013). However, the implementation developed here has a more limited capability due to software limitations.

The cost function returns a scalar which measures the fit of of the model to the observations. The cost function is defined as:

$$J = \gamma_1 \int_{\Gamma_s} w \cdot (U_{obs} - U_s)^2 d\Gamma_s + \gamma_2 \int_{\Gamma_b} (\nabla\alpha \cdot \nabla\alpha) d\Gamma_b \tag{22}$$

where $\gamma_1$ and $\gamma_1$ are user-defined scaling factors, $\Gamma_s$ is the surface domain, $\Gamma_b$ is the basal domain, $w(x,y)$ is a weighting function, $U_{obs}(x,y)$ are observed surface ice speeds, $U_s(x,y)$ are modelled surface speeds, and $\alpha(x,y)$ is the control parameter,

The cost function defined above has two terms: $J = \gamma_1 J_0 + \gamma_2 J_{Reg}$. The first term ($J_0$) measures the weighted square of the difference between observed and modelled velocity. The second term ($J_{Reg}$) is a Tikhonov regularization term, which



penalizes oscillations in $\alpha$ and stabilizes the inversion (Morlighem et al., 2013). Other formulations of the cost function are possible (e.g. Morlighem et al. (2013)).

The weighting function scales the mismatch between the observed and modelled surface velocities. It is used to incorporate a-priori knowledge about the quality of observations. Observations known to greater precision can be weighted higher, such that

they have greater influence on the cost function than observations with a high error. The inverse of the variance of measurements is a statistically desirable weighting function.

The control parameter refers to the variable which the inversion process optimizes in order to best match model prediction and observations. Since our aim is to determine the basal drag, the control parameter is a parameter in the basal sliding law. For the linear sliding law, $\alpha = \beta^2$. For the generalized Weertman sliding law, $\alpha = \mu_a$. Although the Schoof sliding law has

two unknowns which can be inverted for, $\mu_b$ exerts a dominating control. Hence, $\lambda_b$ is set to a constant while $\alpha = \mu_b$. In the numerical implementation of the adjoint, $\alpha$ is parameterized as $\alpha(x,y) = exp(\zeta(x,y))$. This ensures that $\alpha$ remains positive, as expected for each of the three sliding laws. For simplicity, this is neglected in the remainder of the paper, and the discussion focuses on recovering $\alpha$ rather than $\zeta$.

The inversion process aims to determine the field of $\alpha$ which minimizes the cost function. This is an optimization problem.

Starting with an initial guess for $\alpha$, the gradient of the cost function with respect to $\alpha$, is determined. The gradient provides a search direction for the optimization algorithm, which updates $\alpha$. This process is repeated iteratively until $\alpha$ converges below a tolerance or until a maximum number of iterations occur. The critical component in this process is the gradient $\frac{dJ}{d\alpha}$. The process to calculate this gradient is described in the next two subsections.

### 2.2.2  Adjoint model description

The methodology to obtain the gradient $\frac{dJ}{d\alpha}$ follows from Goldberg and Heimbach (2013). The key concepts of this approach are first explained for a generic algorithm, before showing how they can be applied to the ice sheet model. This explanation follows that of Errico (1997) and Goldberg and Heimbach (2013).

Consider the model:

$$b = B(\boldsymbol{\phi}) \tag{23}$$

where $\phi$ is an arbitrary variable (or array of variables), and B can be considered a sequence of operations:

$$B(\boldsymbol{\phi}) = B_N((...(B_2(B_1(B_0(\boldsymbol{\phi}))))))  \tag{24}$$

and each operation can be written as $b_N = B_N()$

Further, define a function J:

$$J = J(b) \tag{25}$$



where J returns a scalar. In the context of the adjoint model, the function is known as the cost function, objective function, or target function (Goldberg and Heimbach, 2013). This function quantifies an aspect of the model output which is of interest, such as the mean error of model output relative to observations.

The aim is to determine the gradient of the cost function J with respect to in the initial input $\phi$. To provide context for the adjoint model, the tangent linear model (TLM) is presented first. In the TLM, a small perturbation in the input is propagated forward through the model to determine the corresponding perturbation in the output. Applying the chain rule to $J = J(b) = J(B(\phi))$ leads to the corresponding TLM:

$$\delta J = \left( \prod_{i=N}^{1} \frac{\partial b_i}{\partial b_{i-1}} \right) \frac{\partial b_0}{\partial \phi_i} \delta \phi_i \tag{26}$$

There are several observations about the TLM. First, the TLM determines the perturbation of $\delta J$ from the perturbation of a single element $\phi_i$. As the perturbation $\delta \phi_i$ approaches zero, $\frac{\delta J}{\delta \phi_i}$ converges to $\frac{dJ}{d\phi_i}$. Second, to determine $\frac{dJ}{d\phi}$, the TLM needs to be run for each entry in $\phi$. Although for small models this approach is feasible, the computational cost is too great for glaciological problems on domains of the size of interest. Finally, the TLM acts in a similar direction as the model B, in that the functions are applied successively starting with the counterpart to $B_0$ (Errico, 1997).

The concept behind the adjoint model is that rather than determining how changes in the input $\phi$ impact the cost function J, it can be more efficient to determine how changes in the cost function J impact the initial input $\phi$. In the adjoint model, sensitivities of J are propagated backwards through the model, to determine the resulting change in $\phi$. Similar to the TLM, the adjoint model is derived by applying the chain rule to $J = J(b) = J(B(\phi))$:

$$\frac{\partial J}{\partial \phi} = \left( \prod_{i=1}^{N} \left[ \frac{\partial b_i}{\partial b_{i-1}} \right]^T \right) \frac{\partial J}{\partial b_N} \tag{27}$$

Key observations about the adjoint model are: 1. In contrast to the TLM, which acts upon a perturbation, the adjoint model acts upon the sensitivity of the cost function. 2. A single run of the adjoint model is sufficient to determine the gradient $\frac{\delta J}{\delta \phi}$. 3. The adjoint model runs in reverse relative to both the model and the TLM, in that the adjoint model applies functions beginning with the counterpart to $B_N$ and ending with the counterpart of $B_0$ (Errico, 1997).

### 2.2.3 Adjoint model implementation

The adjoint model is generated based on automatic differentiation (AD, Griewank and Walther (2008)) of the Matlab code implementations of the forward model. AD tools process an input code to generate a counterpart code which returns the corresponding gradient (or Jacobian). The central concept behind AD is that a computer program is fundamentally a sequence of elementary operations and functions. This admits the repeated application of the chain rule to generate a derivate of high accuracy.

Multiple methodologies exist for AD tools to generate the derivate code. Previous application of AD software to generate the adjoint in glaciology (Heimbach and Bugnion, 2009; Goldberg and Heimbach, 2013; Martin and Monnier, 2014) have





used reverse accumulation AD tools (e.g. Giering et al., 2005; Hascoet and Pascual, 2004). These types of AD software are conceptually similar to the adjoint model. They are designed to determine the gradient of function (input code) by propagating sensitivities of the output variables backwards to the input variables. Hence, an ice sheet model can be processed with relatively little modification by reverse accumulation AD tools to generate the adjoint model.

Here, we apply the open source AD tool ADiGator (Weinstein and Rao, 2011-2016), which in contrast to previous work is a forward accumulation AD tool. The methodology of forward accumulation is conceptually similar to the TLM. It is designed to determine the gradient of a function (input code) by propagating sensitivities of the input variables forward through the program to the output. Applying a forward AD tool to an ice sheet model to generate the adjoint is not feasible due to the size of the control space. Rather, we generate the adjoint by applying ADiGator to segments of the ice sheet model code, and

multiplying the resulting Jacobians following Equation 27.

Pseudocode of the main ice sheet model routine is shown in Algorithm 1, and the corresponding code to calculate the adjoint is shown in Algorithm 2. Two new functions, S1 and S2 appear in the adjoint code. These encapsulate segments of code from the forward model and can be processed by ADiGator. The function S2 contains code which spans over two Picard iterations. The adjoint does not contain a for loop corresponding to iterating through the Picard iterations in reverse (c.f. Goldberg et al.

(2016)). Rather, values from the final two Picard iterations of the forward model are saved and used as input for the adjoint code. The adjoint model is also modified to solve the cubic equation (following Arthern et al. (2015)) to determine $\eta$, rather than storing values from the previous iterations and implementing a fixed point iteration. This impacts the $\eta$, $\bar{\eta}$, and $F_a$ functions, but leaves the overall structure the same. This is a necessary modification for ADiGator.

The adjoint code explicitly calculates several Jacobian matrices (lines 15 to 23 in Algorithm 2). ADiGator is applied to

the corresponding functions to generate the Jacobian matrices, except the solution to the system of linear equations, which requires special treatment. A counterpart to the linear solve which returns the corresponding derivate is manually programmed following the procedure detailed in the appendix of Martin and Monnier (2014). The adjoint is then calculated by multiplying out the sensitivities of the cost function with the transposes of the Jacobian matrices. Although this process is more complicated and less flexible than previous approaches, it is necessary as no non-commercial AD reverse accumulation tool is available for

Matlab.

This implementation of the adjoint is equivalent to previously published adjoint implementations (Goldberg and Heimbach, 2013; Martin and Monnier, 2014) restricted to one reverse step in the Picard iteration. This is mathematically equivalent to the Lagrangian Multiplier method introduced by MacAyeal (1993) (Heimbach and Bugnion, 2009).

The gradient from the adjoint model is used to solve the optimization problem which minimizes the cost function. The

inversion code relies on minFunc (Schmidt, 2005), a publicly available Matlab unconstrained optimization package. The L-BFGS routine, with a Wolfe Condition backtracking line search, is applied in the inversion code. The cost function is discretized using the same finite difference operators as the ice sheet model.

Performance of the inversion code was verified using a series of identical twin-tests (Goldberg and Heimbach, 2013). Results are shown in Koziol (2017).



---

**Algorithm 1** Ice sheet model main routine pseudocode

---

1:  Initialize: $\bar{u}, \eta, C, C_{eff}, \alpha$     %From Shallow Ice Approximation

2:

3:  for $j = 1, 2, 3, ..., N$     %Picard Iterations

4:     $\eta = \boldsymbol{\eta}(\bar{u}, \eta, C_{eff})$     %Viscosity (Eq. 9)

5:     $\bar{\eta} = \bar{\boldsymbol{\eta}}(\eta)$     %Depth integrated Viscosity (Eq. 20)

6:     $F_2 = \boldsymbol{F_a}(\eta, a = 2)$     %F-integral (Eq. 13)

7:     $C = \boldsymbol{C}(\bar{u}, \alpha, C, F2)$     %Basal drag parameter (Eq. 6)

8:     $C_{eff} = \boldsymbol{C_{eff}}(C, F2)$     %Effective basal drag parameter (Eq. 16)

9:

10:     $\bar{u} = \boldsymbol{u}(C_{eff}, \bar{\eta})$     %Velocities (Eq. 21)

11:  end

12:

13:  $J = \boldsymbol{J}(\bar{u}, C_{eff}, \alpha)$     %Cost Function Eq. 27

---

## 2.3   Subglacial Hydrology Model

This subglacial hydrology model used is described in detail in Hewitt (2013) and Banwell et al. (2016), and similar conceptually to the model presented in Werder et al. (2013). Here, the version employed in Banwell et al. (2016) is applied.

Both distributed and channelized flow are represented in the subglacial hydrology model. Distributed flow is described by an average thickness and flux over a representative area. As in Banwell et al. (2016), the distributed system is composed into two components: a cavity sheet, and an elastic sheet. The elastic sheet is included to simulate 'hydraulic jacking' from lake hydrofracture events, and is activated only when the effective pressure drops to zero and below. Channels have the potential to form along the edges and diagonals of the numerical grid. Channels are initiated by dissipative heating from the distributed system over an incipient channel width lengthscale. The model is written in Matlab, using a finite difference numerical grid, and an implicit forward time step method. For full details, consult Hewitt (2013) and Banwell et al. (2016).

## 2.4   Application to Russell Glacier Area

The Russell Glacier area is a land-terminating sector of the Greenland Ice Sheet (Figure 1). The ice sheet model and inversion code are applied to the Russell Glacier area to determine the basal boundary condition at the end of the winter season.

An outline of the study area is shown in (Figure 1). The northern and southern boundaries are selected to be roughly in line with basal watersheds determined using the Shreve (1972) approximation for hydraulic gradient. The northern boundary is approximately the same as used by Bougamont et al. (2014) and de Fleurian et al. (2016). The southern boundary is further south relative to Bougamont et al. (2014), but north of the southern boundary in de Fleurian et al. (2016). The eastern boundary was selected to extend up ice of the GPS stations (Tedstone and Neinow, 2017). The western boundary is the ice-margin. There is a nunatak near the western boundary.




---

**Algorithm 2** Pseudocode of the adjoint code

---

1: % Encapsulate segments of code into functions

2: **function** $S1(\bar{u}, \alpha, C, F2)$

3: $C = \boldsymbol{C}(\bar{u}, \alpha, C, F2)$

4: $C_{eff} = \boldsymbol{C_{eff}}(C, F2)$

5: **return** $C_{eff}$

6:

7: **function** $S2(\bar{u}, \alpha, C, F2)$

8: $C = \boldsymbol{C}(\bar{u}, \alpha, C, F2)$

9: $C_{eff} = \boldsymbol{C_{eff}}(C, F2)$

10: $\eta = \boldsymbol{\eta}(\bar{u}, C_{eff})$

11: $\bar{\eta} = \boldsymbol{\bar{\eta}}(\eta)$

12: **return** $\bar{\eta}$

13:

14: % Calculate Jacobian matrices $(\mathrm{D}f|_p = \frac{\partial f_i}{\partial p_j})$

15: $\mathrm{D}J|_{\bar{u}} = \textbf{ADiGator}(\boldsymbol{J}, \bar{u}^N, C_{eff}^N, \alpha)$

16: $\mathrm{D}J|_{C_{eff}} = \textbf{ADiGator}(\boldsymbol{J}, \bar{u}^N, C_{eff}^N, \alpha)$

17: $\mathrm{D}J|_{\alpha} = \textbf{ADiGator}(\boldsymbol{J}, \bar{u}^N, C_{eff}^N, \alpha)$

18:

19: $\mathrm{D}S1|_{\alpha} = \textbf{ADiGator}(\textbf{S1}, \bar{u}^{N-1}, \alpha, C^{N-1}, F_2^N)$

20: $\mathrm{D}S2|_{\alpha} = \textbf{ADiGator}(\textbf{S2}, \bar{u}^{N-1}, \alpha, C^{N-2}, F_2^{N-1})$

21:

22: % The Jacobian of the velocity solve is calculated using a manually programmed function

23: $\mathrm{D}u|_{C_{eff}} = \textbf{U\_jac}(\bar{u}^N, C_{eff}^N, \bar{\eta})$

24: $\mathrm{D}u|_{\bar{\eta}} = \textbf{U\_jac}(\bar{u}^N, C_{eff}^N, \bar{\eta})$

25:

26: % The adjoint (Eq. 27)

27: $\frac{dJ}{d\alpha} = (\mathrm{D}S1|_{\alpha})^T (\mathrm{D}\bar{u}|_{C_{eff}})^T (\mathrm{D}J|_{\bar{u}}) +$

28: $\qquad (\mathrm{D}S2|_{\alpha})^T (\mathrm{D}\bar{u}|_{\bar{\eta}})^T (\mathrm{D}J|_{\bar{u}}) +$

29: $\qquad (\mathrm{D}S1|_{\alpha})^T (\mathrm{D}J|_{C_{eff}}) + \mathrm{D}J|_{\alpha}$

---





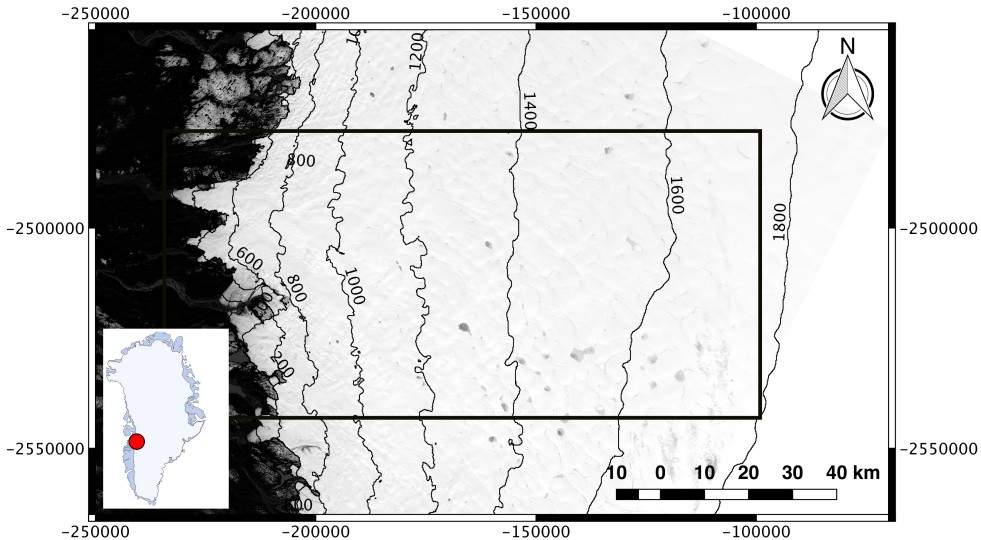

**Figure 1.** Landsat 8 satellite image, band 2, showing the Russell Glacier area. Black box outlines the study area. Inset shows the location in reference to Greenland.

The ice sheet model/inversion code are applied to determine the basal boundary condition at the end of the 2008-2009 winter season in the Russell Glacier study site. The end of the winter season is assumed to be day 120 of the year (April 30th). Although the exact day is somewhat arbitrary, this day was selected as it is shortly before surface runoff begins in the study area, and shortly before GPS records in the study site show enhanced motion (Tedstone and Neinow, 2017).

Applying the ice sheet model/inversion code to the Russell Glacier area requires a number of datasets. Mean winter surface velocities for 2008/2009 (Figure 2) are provided by the MEaSUREs Greenland Ice Sheet Velocity Map at 500 m resolution (Joughin et al., 2010a, b). Surface and basal topography (Figure 3) are provided by the BedMachine2 dataset (Morlighem et al., 2014, 2015), and are resampled to 500 m resolution from 150 m resolution to match the velocity data. This is slightly coarser than the reported true resolution of 400 m for the ice thickness. The 500m grid resolution results in a grid size of 132x274 for

the domain. Fifty vertical layers are used for integration using Simpson's rule.

An important assumption made is that the mean winter velocities are representative of both the beginning and end of winter. This assumption is justified by observing published GPS records in Southwest Greenland (Colgan et al., 2012; van de Wal et al., 2015). These observations show that although velocities increase throughout the winter, the magnitude of the change is relatively limited.

Inversions are initialized using a basal drag set to the local driving stress smoothed by a 3x3 grid cell mean filter. The ice-margin boundary is described in the ice sheet model by Equation 7 and 8 while on the three other boundaries a Dirichlet boundary condition is applied. The inverse of the errors provided with the surface velocity measurements are used as weights in the cost function.





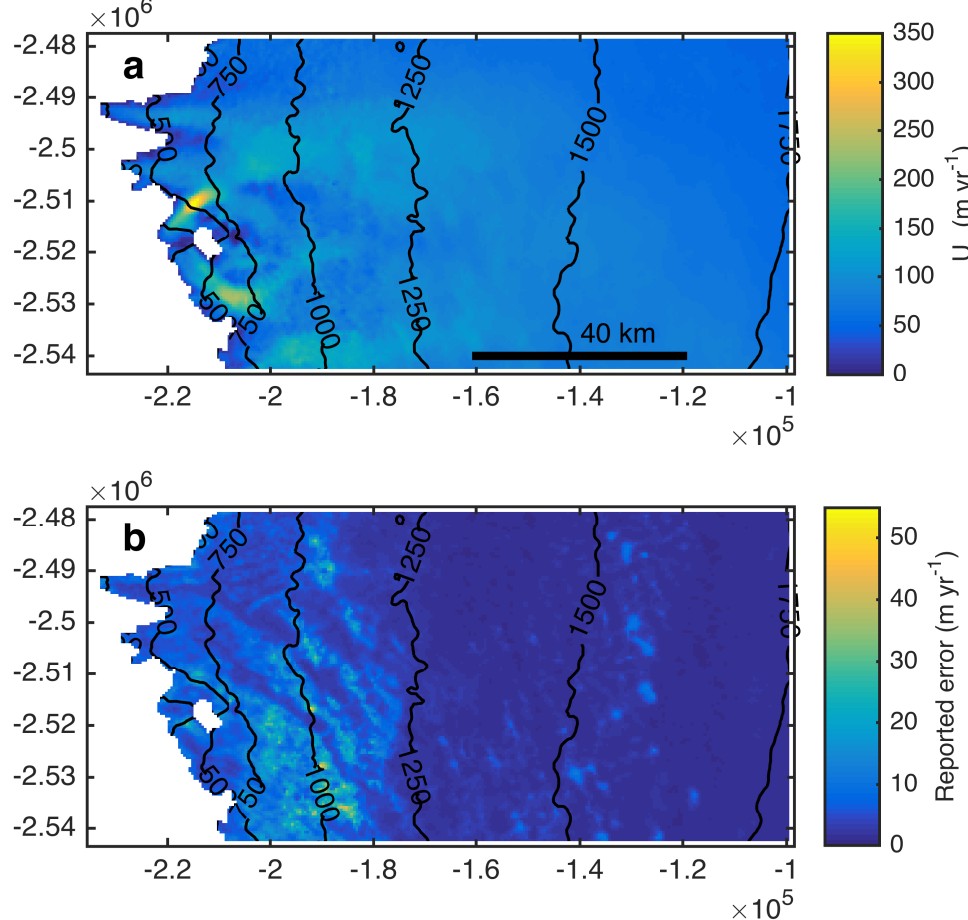

**Figure 2.** a) Velocity measurements from the MEaSUREs Greenland Ice Sheet Velocity Map at 500 m resolution for the Russell Glacier area (Joughin et al., 2010a, b). b) Reported error for the measurements.

The results of inversions depends on the relative values of the scaling factors $\gamma_1$ to $\gamma_2$ in the cost function (Equation 22). For each sliding law, a series of inversions is performed with $\gamma_1$ set to 1 while varying $\gamma_2$. A L-curve analysis is applied to select the inversion which best balances fitting the velocity observations while penalizing spurious oscillations in basal drag.

Parameters for the ice sheet model/inversion code are listed in Table 1. Similar to Hewitt (2013), the ice flow creep parameter ($A$) is selected to be $7 \cdot 10^{-25}$ Pa$^3$s. This corresponds to an ice temperature of approximately -7 °(Cuffey and Paterson, 2010). This choice for $A$ results in the ratio of basal velocity to surface velocity remaining greater than 0.5 throughout the study area.

The parameters for the subglacial hydrology are the result of an extensive parameter search using a coupled ice-flow/subglacial hydrology model in (Koziol, 2017). Many of the parameters are the same as published in Banwell et al. (2016) and Hewitt (2013). However, testing of the reported optimal parameters for the Paakitsoq region reported by Banwell et al. (2016) using





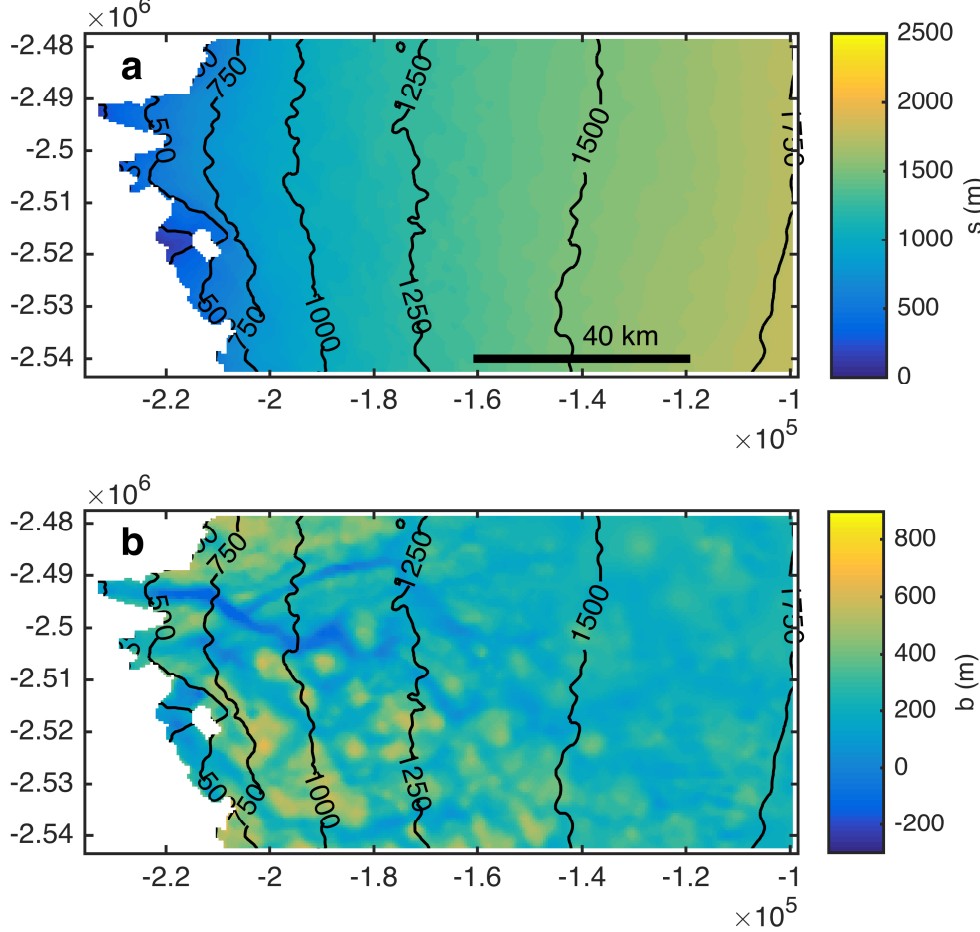

**Figure 3.** a) Surface topography from BedMachine2 dataset (Morlighem et al., 2014, 2015) reinterpolated to 500 m. b) Basal topography at same resolution.

the integrated model showed poor agreement with GPS measurements due to insufficient volumes of water being evacuated from mid-elevations.

The workflow developed for incorporating modelled effective pressure into inversions using non-linear sliding laws is show in Figure 4. This workflow is motivated by the idea that both the subglacial hydrological system and ice flow are in quasi-steady state during the winter. This allows us to invert for background values of the constants in the sliding laws. The initial step is to invert using a linear sliding law for the basal drag coefficient. Basal velocities are calculated from modelled depth integrated velocities (Equation 15). The modelled basal drag and basal velocities then provide the necessary input for the subglacial hydrology model to calculate a distributed basal melt rate. The modelled distributed basal melt rate incorporates geothermal flux, but neglects heat loss to the interior of the ice sheet (Hewitt, 2013).





| Symbol | Constant | Value | Units |
|---|---|---|---|
| A | Ice-flow parameter | $7 \cdot 10^{-25}$ | $\mathrm{Pa^{n}s^{-1}}$ |
| $A_b$ | Ice-flow parameter for basal ice | $7 \cdot 10^{-24}$ | $\mathrm{Pa^{n}s^{-1}}$ |
| $\rho_i$ | Ice density | 917 | $\mathrm{kgm^{-3}}$ |
| g | Gravitational constant | 9.81 | $\mathrm{ms^{-2}}$ |
| n | Exponent in Glen's flow law | 3 | |
| p | Exponent generalized Weertman sliding law | $3^{-1}$ | |
| q | Exponent generalized Weertman sliding law | $3^{-1}$ | |
| $\lambda_b$ | bed roughness scale | 1 | m |
| $t_y$ | Seconds per year | 31536000 | $\mathrm{syr^{-1}}$ |
| $\epsilon$ | viscosity regularization parameter | $1 \cdot 10^{-14}$ | $\mathrm{ms^{-1}}$ |

**Table 1.** Constants used in the ice sheet/inversion model applied to the Russell Glacier area.

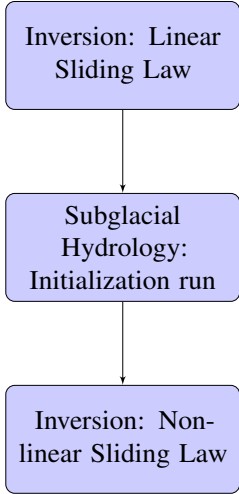

**Figure 4.** Flow chart showing the work flow for non-linear inversions





The subglacial hydrology model is then run for the winter season with the basal drag and basal velocities from the linear inversion. The model is run at 500m resolution (identical to the inversions), with no-flow boundary conditions at the northern, southern, and eastern boundaries. The ice-margin is assumed to be at atmospheric pressure. This boundary condition is modified at necessary places to prevent inflow of water from beyond the ice sheet margin. Similarly to Banwell et al. (2016), the

subglacial hydrology model is initialized with the thickness of the sheet flow layer set to 0.10 m. Testing showed that varying initial thickness has negligible impact. At this stage, the ice sheet model remains unconnected, and the input basal velocities are assumed to be constant. The subglacial hydrology model run provides a modelled water pressure distribution over the study site.

Finally, the non-linear inversions are run using the modelled water pressure from the subglacial hydrology model winter run.

Two sets of inversions are conducted, one for the generalized Weertman sliding law, and one for the Schoof sliding law. The first set of inversions seeks to determine the distribution of $\mu_a$, while the second inverts for $\mu_b$. Similar to the linear sliding law, an L-curve analysis is employed to determine the relative values of $\gamma_1$ to $\gamma_2$.

## 3   Results

### 3.1   Linear Inversion

Six inversions using the linear sliding law are run (Figure 5). Using the L-curve plot, the inversion with $\gamma_2 = 1 \cdot 10^{-12}$ is selected as optimal. The value of $J_0$ for this inversion is $1.56 \cdot 10^{11}$.

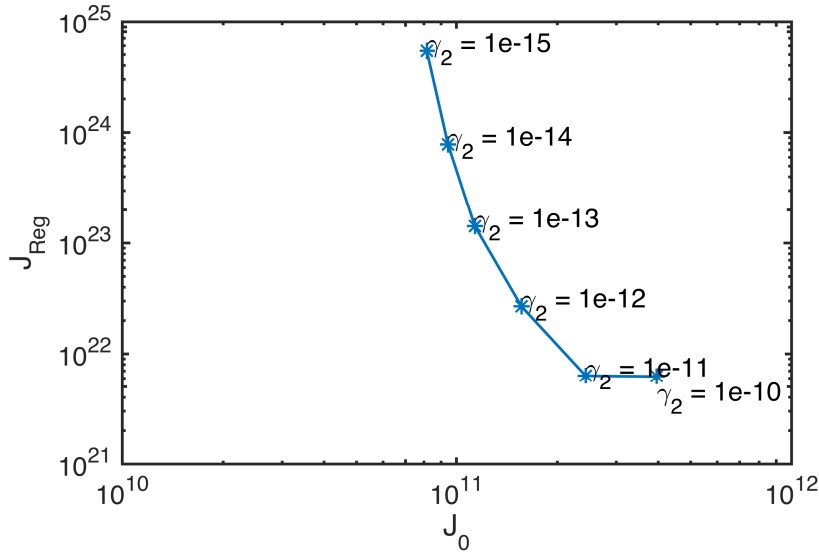

**Figure 5.** Log-log plot for L-curve analysis of inversions of the Russell Glacier area employing a linear sliding law.

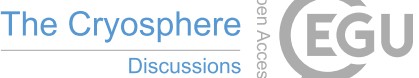

The inversion converges in 46 iterations (Figure 6). The histogram of the the difference between observed and modelled velocities (Figure 7) has a maximum in the lowest bin, with a rapid decrease into a long tail. The maximum difference is approximately 165 $\mathrm{myr}^{-1}$. The difference between modelled and observed velocities is less than 10 $\mathrm{myr}^{-1}$ for 88% of the cells in the study area, and less than 20 $\mathrm{myr}^{-1}$ for 96% of the cells in the study area. A map of the difference between observed and modelled velocities shows the highest difference occurs along the ice-margin and in the vicinity of the nunatak (Figure 7b). Figure 8 shows the inverted basal drag parameter, basal drag, and the sliding ratio for the linear sliding law.

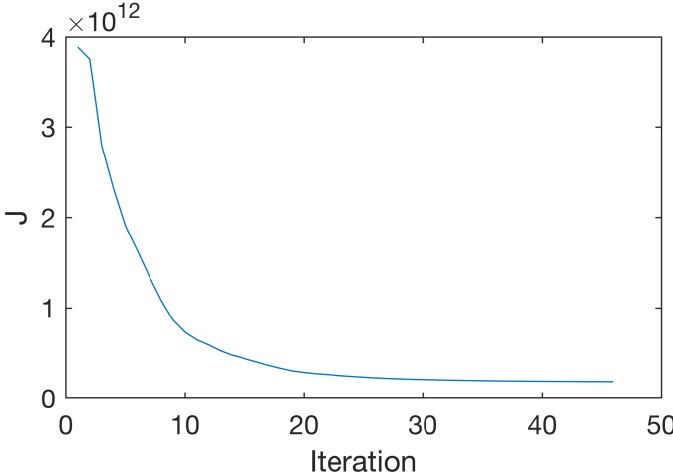

**Figure 6.** Convergence of the optimization routine for the inversion using a linear sliding law.

## 3.2   Subglacial Hydrology Model

Basal melt during the winter is shown in Figure 9. Most values are between 0.015 and 0.03 $\mathrm{myr}^{-1}$, with higher values predominately occurring near the nunatak. The spatial pattern of melt broadly reflects the patterns of surface velocities (Figure 2).

The subglacial hydrology model winter run evolves rapidly at the beginning of the run (Figure 10). By day 50 of the model run, the rate of change is significantly reduced. At day 240 of the run the model is in an approximate steady state. Relative to discharge at the base, changes in effective pressures have a much lower magnitude.

The distribution of sheet thickness at the end of winter mirrors basal topography, with the sheet thickest in topographic lows (Figure 11). The maximum sheet thickness is 0.36 m, which is less than the bed roughness scale of 0.5 m. The effective pressure also reflects the basal topography, with lowest effective pressures located in topographic lows. Since the lowest effective pressure is 0.44 mPa, no part of the ice sheet is near flotation. The model predicts minor channelization in two locations (not shown), with single channels extending from the margin several kilometres.





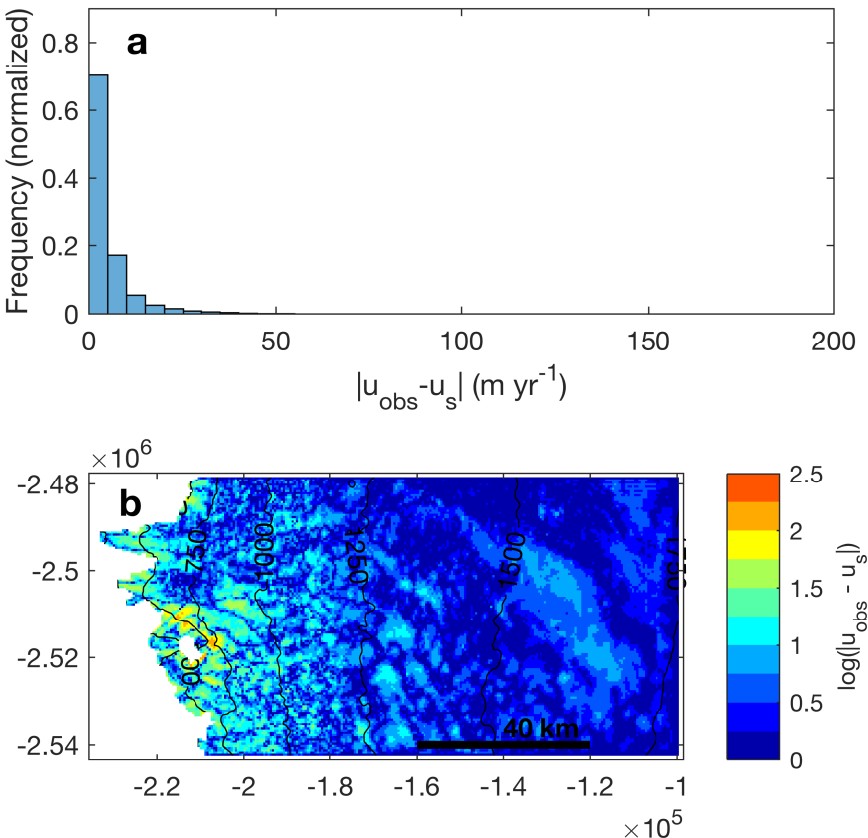

**Figure 7.** a) Histogram of the absolute difference between the observed and modelled surface velocities for the inversion using a linear sliding law. b) Map of the log of the absolute difference between the observed and modelled surface velocities for the same inversion.

### 3.3 Non-linear sliding laws

An L-curve analysis (Figure 12 and 15) is used to determine the optimum inversion for each of the non-linear sliding laws. The inversions corresponding to $\gamma_2 = 1$ is selected for the Weertman sliding law, while the inversion corresponding to $\gamma_2 = 10^{11}$ is selected for the Schoof sliding law. These were selected so that the cost term of the inversions were similar to that of the linear

5 sliding law. The two cost terms for the Weertman and Schoof sliding laws are $J_0 = 1.78 \cdot 10^{11}$ and $J_0 = 1.60 \cdot 10^{11}$ respectively.

The histogram of the absolute difference between observed and modelled surface velocities for both non-linear sliding laws shows a similar distribution to the linear sliding law inversion (Figure 13 and 16). The Weertman sliding law results in a spatial distribution of misfit similar to the linear sliding law, while spatial distribution of error from the Schoof sliding law shows higher frequency variations. Model mismatch again is highest in the vicinity of the nunatak.

10 Figure (14 and 17) show the inversion results from the Weertman and Schoof sliding law respectively. Inverted basal drag using the Weertman sliding law is very similar to the results from the linear sliding law. In contrast, the inverted basal drag



**Figure 8.** Inversion results using the linear sliding law. a) Inverted drag parameter. b) Basal drag. c) Sliding ratio.

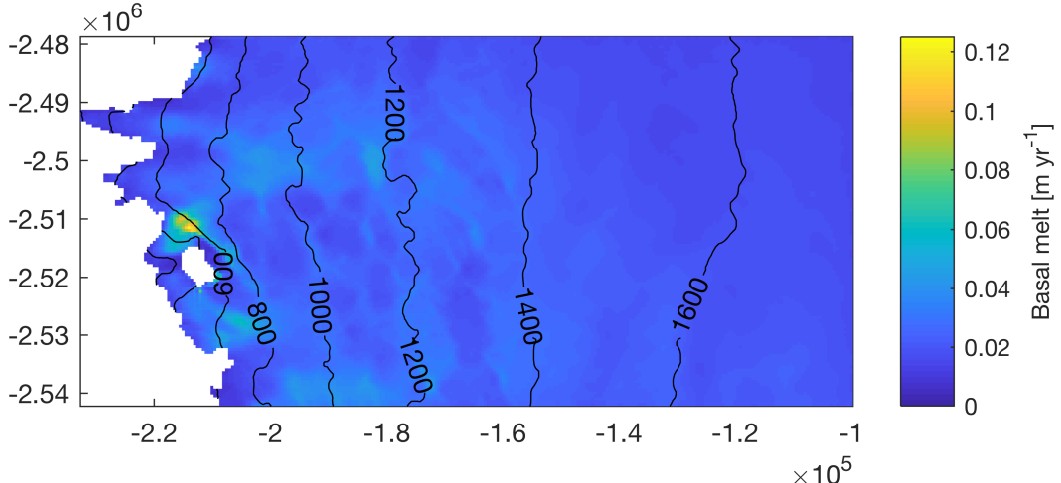

**Figure 9.** Modelled basal melt rate using basal velocities from linear inversion.

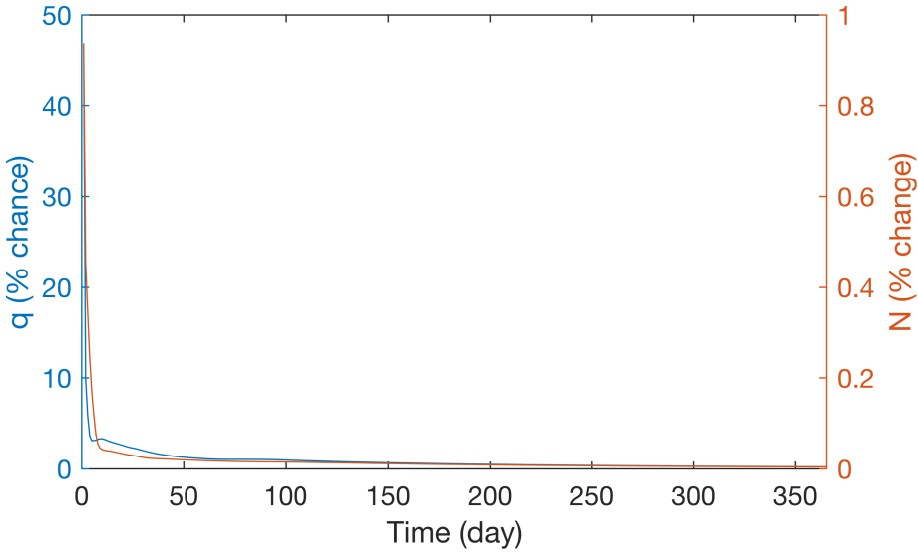

**Figure 10.** Plot showing the mean daily % change in water flux and effective pressure for a one year run.





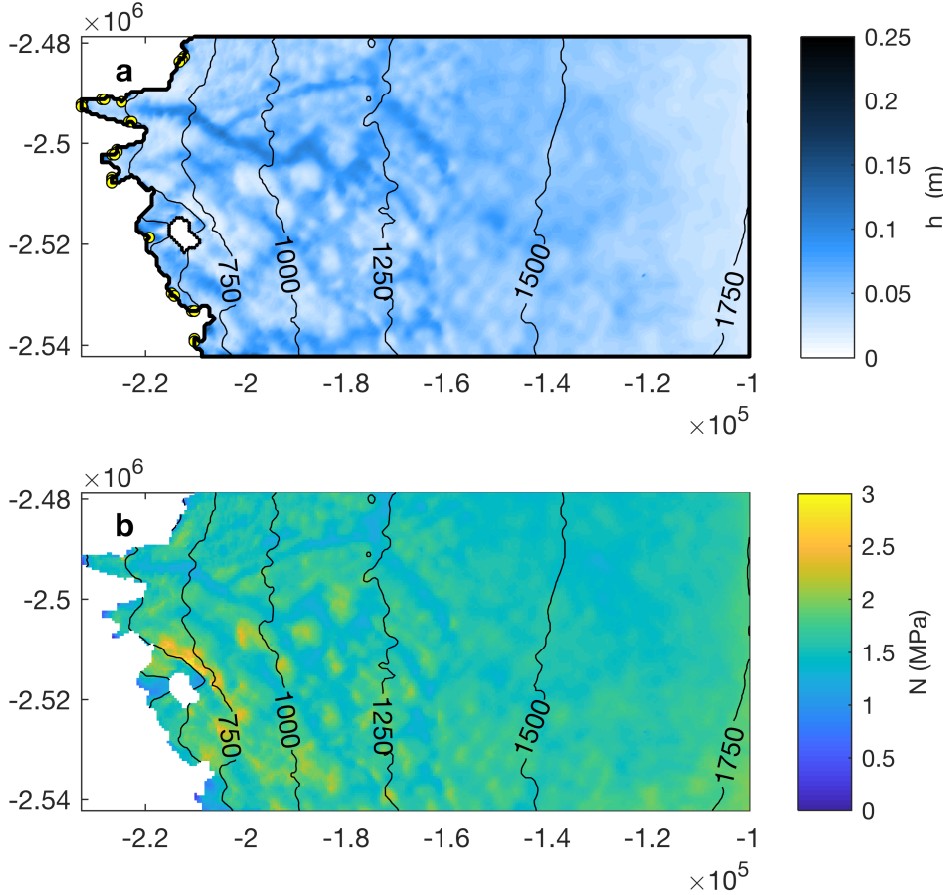

**Figure 11.** Modelled state of the subglacial hydrology system at the end of the winter. a) Map of sheet thickness, with black contours showing surface elevation. b) Map of effective pressure overlaid with surface elevation contours.

from the Schoof sliding law shows much higher frequency and magnitude spatial variations. This is reflected in the spatial distribution of the sliding ratio, with the Weertman sliding law resulting in a distribution similar to the linear sliding law, while the distribution from the Schoof sliding law shows much greater variation.

## 4  Discussion

Inversions of the Russell Glacier area are run with a constant creep parameter $A$, corresponding to an ice temperature of approximately -7 °C (Cuffey and Paterson, 2010). For inversions with the linear sliding law, tests showed poorer results when $A$ increased (corresponding to warmer ice). As $A$ decreased, the sliding ratio approached one uniformly, and inversion results were better able to match observed surface velocities. The value of $A$ was selected as a balance of model fit, while keeping a contribution to motion from internal deformation. Observations from two boreholes located in the Paakitsoq region show that





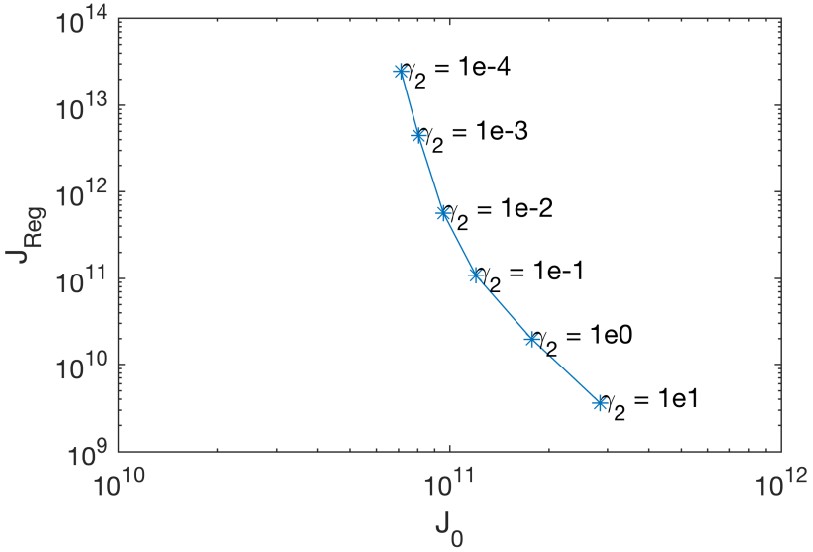

**Figure 12.** Log-log plot for L-curve analysis of inversions of the Russell Glacier area employing generalized Weertman sliding law.

internal deformation results in approximately 27-56% of ice velocities during winter (Ryser et al., 2014). In reality, $A$ would have a heterogeneous distribution. By using a constant $A$, the basal drag parameter will account for some of the effects which would otherwise be due to variation in $A$.

Basal velocities determined from the optimal inversion using a linear sliding law are input into the subglacial hydrology

model. The distribution of basal velocities is used to both calculate the basal melt rate, and the cavity space in the continuum sheet flow. Due to the selection of a creep parameter such that the sliding ratio is relatively high, it is likely that basal velocities are overestimated. This would result in an overestimate of water generated at the ice-bed interface, and an overestimate of the capacity of cavity space. Application of a higher order ice sheet model would be advantageous in these regards.

The pattern of basal drag inverted using the linear and Weertman sliding law show limited differences. This is due to the

fact that basal shear traction must satisfy the global stress balance (Joughin et al., 2004; Minchew et al., 2016). Both the linear and Weertman sliding law have the form $\tau_b = C \cdot u^{1/m}$ in the inversion, since effective pressure can be incorporated into the constant C for the Weertman sliding law. Previous work shows that in this case $C \propto u^{-1/m}$, and the recovered fields of basal drag are within a few percent of each other (Minchew et al., 2016). The basal drag and basal velocities from the the linear sliding law to initiate the subglacial hydrology model are therefore self consistent with the subsequent inversion results of

the Weertman sliding law. The pattern of basal drag inverted using the Schoof sliding law however, shows both higher spatial variability and a higher magnitude of variability. This is a result of the Schoof sliding law shifting to Coulumb-like behaviour at low effective pressures.

Interpretation of radar lines in the Russell Glacier area suggests significant winter storage of water along topographic highs, while significant water flow through topographic lows occurs during the summer melt seasons (Chu et al., 2016). Based on





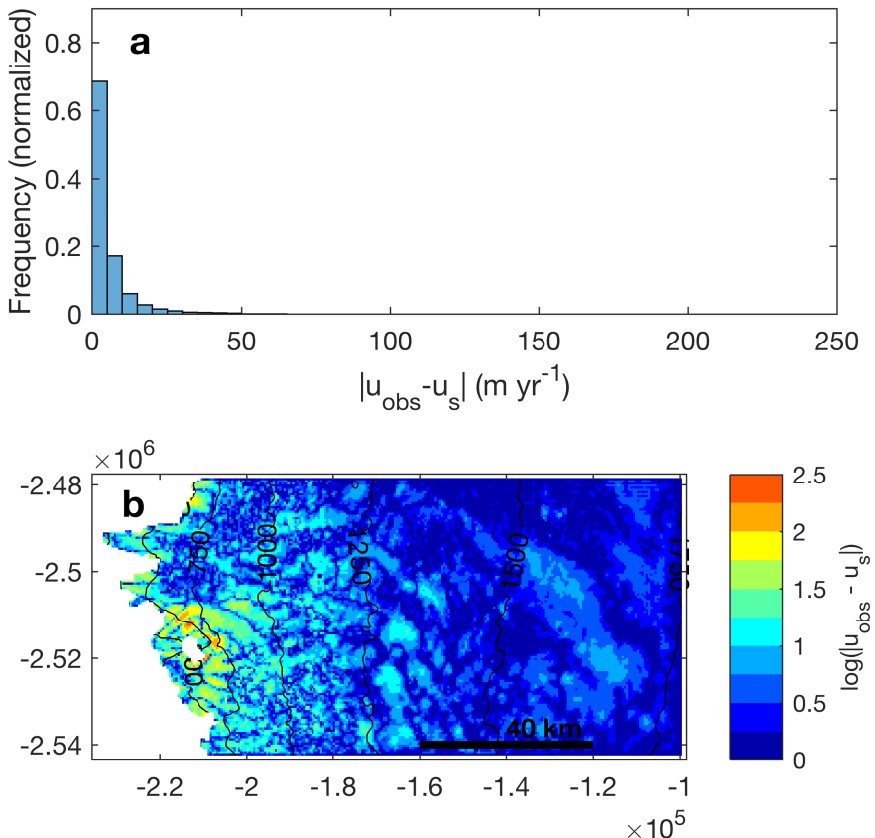

**Figure 13.** a) Histogram of the absolute difference between the observed and modelled surface velocities for the inversion using a generalized Weertman sliding law. b) Map of the log of the absolute difference between the observed and modelled surface velocities for the same inversion

these observations, the subglacial hydrology runs are is reflective of summer conditions rather than winter conditions. Water storage, which would be characterized by high sheet thickness, is not observed along topographic highs. Chu et al. (2016) attribute storage on topographic ridges to water storage in parts of the distributed system which become isolated at the end of the melt season. In contrast, porous sediments in bedrock troughs are hypothesized to allow water to drain (Chu et al., 2016).

5   The treatment of the bed in the subglacial hydrology model is uniform. It does not account for differences in till cover or bed properties, nor does it account for sub-grid scale heterogeneity in the distributed system, which is likely the cause of water storage. Replicating these observations likely requires the implementation of another model component, such as the weakly connected distributed system proposed by Hoffman et al. (2016). In general, model output from the subglacial hydrology model can be expected to be much more sensitive to the model formulation during the winter than the summer, when the system is

10   forced by high water input. Inline with inferences from tracer injections (Chandler et al., 2013), the model does not predict a channelized system at the margin during the winter.





**Figure 14.** Inversion results using the Weertman sliding law. a) Inverted drag parameter. b) Basal drag. c) Sliding ratio.





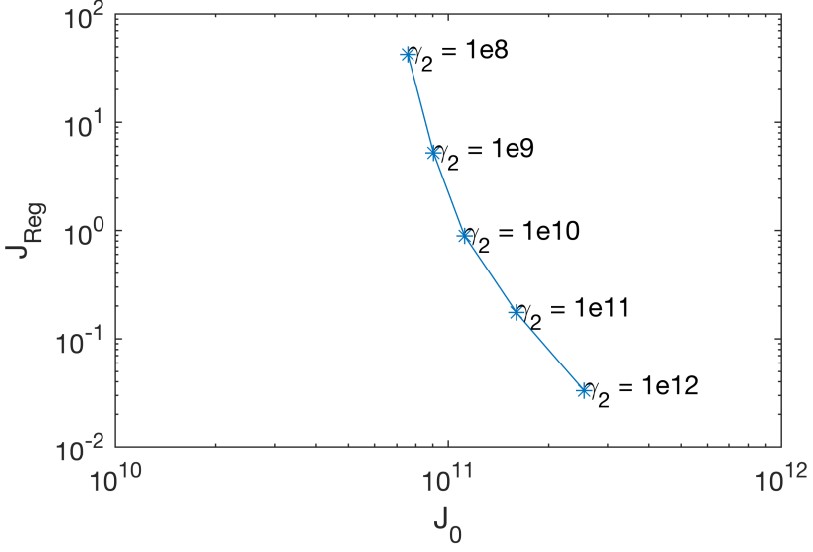

**Figure 15.** Log-log plot for L-curve analysis of inversions of the Russell Glacier area employing the Schoof sliding law.

The initialization procedure introduced is not capable of producing the inferred year on year differences in the subglacial hydrological system at the end of winter. Currently, the subglacial hydrology reaches an approximate steady state by day 240, and is not particularly sensitive to the initialization of the distributed sheet thickness. A full steady state takes approximately two years (Hewitt, 2013). In contrast, observations suggest that summer melt has an impact on the state of the hydrological

system during the subsequent winter (Chu et al., 2016; Sole et al., 2013). The model output of the model therefore can only be considered an approximation to a generic hydrological state. Any discrepancy between the modelled and actual hydrological system is expected to have a greater impact on inversions using the Schoof sliding law, since it has a stronger dependence on effective pressure. In the limit of viscous flow, the Schoof sliding law depends on N. In contrast, the generalized Weertman law is a function of $N^{1/3}$ (Budd et al., 1979). All inversions are conducted using mean winter velocities from 2008-2009. Annual

differences in mean winter velocities are expected to have a minimal impact, as observed year on year differences are on the order 20 $\mathrm{myr}^{-1}$, which is not significantly greater than the velocity mismatch in the inversions.

Other procedures for determining the background parameters of sliding laws can likely be devised. Currently the procedure only uses mean winter velocities. Using mean annual velocities may improve estimates of the sliding law parameters by incorporating information from the melt season. A subglacial hydrological model could be run for an entire year, and basal

parameters determined from an annual average water pressure. A key difficulty is running the hydrological model during the summer, as the development of the system is known to depend on feedbacks with velocity (Hoffman and Price, 2014). This issue can be avoided by using velocity measurements from remote-sensing as a model forcing (e.g Fahnestock et al., 2016). An advantage of running the subglacial hydrology model during the summer months is that model output may be more representative of water flow beneath the ice sheet. Although in its current form the model is too complex, a simplified



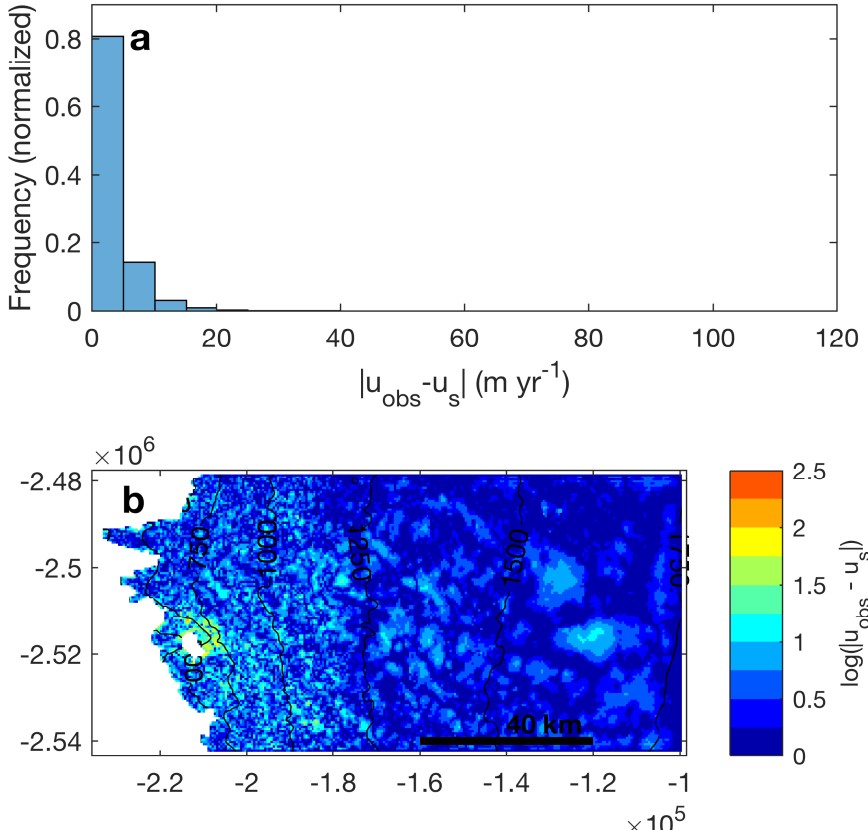

**Figure 16.** a) Histogram of the absolute difference between the observed and modelled surface velocities for the inversion using a Schoof sliding law. b) Map of the log of the absolute difference between the observed and modelled surface velocities for the same inversion.

subglacial hydrology model may be suitable to time dependent adjoint modelling (Goldberg and Heimbach, 2013). Here, we have assumed that the parameters of the sliding law are time independent. This assumption is better suited for bedrock than till, as properties of till are dependent on saturation and deformational history (Minchew et al., 2016).

## 5 Conclusions

5    A new ice sheet model and adjoint code are presented. The ice sheet model is coupled to a recent subglacial hydrology model (Hewitt, 2013). A procedure for initializing a coupled subglacial hydrology/ice sheet model using a winter run is also proposed. The modelled state of the subglacial hydrological system at the end of winter appears to reflect summer observations rather than winter observations. This is likely the result of model formulation rather than the initialization procedure, and the initialization procedure should continue to prove useful as model development advances. The results are subsequently used to run inversions

10    using non-linear sliding laws which are functions of effective pressure. This allows the background parameters for the sliding





**Figure 17.** Inversion results using the Schoof sliding law. a) Inverted drag parameter. b) Basal drag. c) Sliding ratio.





law to be determined. To date, this appears to be the first work to incorporate modelled water pressures in an inversion, and the first to invert with a sliding law explicitly dependent on effective pressure. The usefulness of this inversion for initiating coupled ice sheet/ hydrology model simulations is shown in an upcoming publication.

*Code and data availability.* All datasets used are publicly available. Code is currently not available.

5 *Competing interests.* The authors declare that they have no conflict of interest.

*Acknowledgements.* We would like to thank the M. Morlighem and I. Joughin for the BedMachine2 and MEaSUREs datasets, and I. Hewitt for generously sharing the subglacial hydrology code. C.P. Koziol would like to acknowledge R. Arthern for guidance on writing an ice sheet model and inversion code, as well as B. Minchew, P. Christoffersen, and D. Goldberg for thoughtful discussions. C.P. Koziol was funded through St. John's College, Cambridge and (in part) by NERC standard grant NE/M003590/1.



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
