# Peer review of "Incorporating modelled subglacial hydrology into inversions for basal drag"

_The Cryosphere, 2017_

## Referee Comment (RC1) · Anonymous Referee #1 · 30 Aug 2017

This manuscript describes how to incorporate the effective pressure computed by a subglacial hydrology model into a basal drag inversion. Three different sliding laws are used: (1) a linear sliding law that does not depend on the effective pressure, (2) a "Budd"-type sliding law and a (3) a Schoof sliding law that both depend on the effective pressure. The authors propose here to do an inversion based on the first sliding law (1), to get a good initial guess of basal sliding, then run a subglacial hydrology model for the winter season, which computes the basal melt rate and basal water pressure that is then used to invert for the basal friction parameters used by the two other laws (2) and (3). They find that the basal drag and sliding ratio (ratio between the surface and basal velocities) of the linear and Budd sliding laws are in good agreement, but that their results using the Schoof sliding law show higher spatial variability.

[Figure]

Overall, this manuscript is easy to read and the methods are explained in detail, so that this manuscript is accessible to readers that are not necessarily familiar with model inversions. While I enjoyed reading it, I would have liked to see more discussion on their results instead of focusing mainly on the technical aspect. In its current state, the manuscript is limited to the introduction of a new model and a new approach, which would be more suitable to GMD for example. I do think that there is potential here for more scientific discussion. I am also a bit puzzled by their results with the third sliding law (see below), and why the slip ratios are so different. Since the same viscosity parameters are used for all three models, if the basal velocities are the same, the surface velocities should also be the same. Since we are trying here to reduce the misfit between InSAR velocities and modeled surface velocities, I would expect to see the same basal sliding velocities in all three cases, and this is not what is found here.

**1   Major comments**

I have one main concern, I don't understand the results of the Schoof sliding law. I read several times the explanations (page 22), but I still don't understand why the results are so different.

First, and maybe I am wrong, I don't think we need to do a second inversion once we have a good estimate of basal drag ($\tau_b$) and basal velocities ($u_b$) because there are always ways to change the friction parameters of different sliding laws to end up with the same basal velocities. If this is achieved, then the surface velocities are the same since the internal deformation is the same irrespective of the sliding law. In other words, one can invert for basal friction using a linear sliding law, and then use the results of a subglacial hydrology model to constrain the parameters of a different law using the results of the first inversion without performing another inversion.

For example, here, the three sliding laws are:

$$\tau_b = -\beta^2 u_b \tag{1}$$

$$\tau_b = -\mu_a N_+^p U_b^q \frac{u_b}{U_b} \tag{2}$$

$$\tau_b = -\mu_b N_+ \left( \frac{U_b}{U_b + \lambda_b A_b N_+^n} \right)^{1/n} \frac{u_b}{U_b} \tag{3}$$

and the authors invert for $\beta$ in (1), $\mu_a$ in (2) and $\mu_b$ in (3). If we invert for $\beta$ and compute $U_b$ and $N_+$, one can determine $\mu_a$ and $\mu_b$ by simply doing:

$$\mu_a = \frac{\beta^2 U_b}{N_+^p U_b^q} \tag{4}$$

and

$$\mu_b = \frac{\beta^2 U_b}{N_+ \left( \frac{U_b}{U_b + \lambda_b A_b N_+^n} \right)^{1/n}} \tag{5}$$

Using these values for $\mu_a$ and $\mu_b$, the forward model should produce the same surface velocities and therefore the same misfit to observations. The only problem might be the smoothness of these fields. So, I guess I have 2 questions:

- Why do we need such a complicated procedure, when a simple single inversion would potentially enough?

- Why are the results of the Schoof sliding law so different? Is it because the inversion converged in a local minimum?

It would be nice to try and start with the $\mu_a$ and $\mu_b$ from equations 4 and 5 and see if you indeed get the same sliding ratio for all 3 sliding laws.

**2  Specific comments**

- p3 equations 3, 4 and 5: I think you are missing a minus sign for all these equations (basal drag opposes motion)

- p3 eq 5: use `\left( \right)` rather than simple parentheses.

- p3 l20: I would rather call this equation a Budd sliding law since he is the one who introduced effective pressure in basal stress.

- p3 l24: you should take the norm of $\tau_b$ here (not the vector) since you are comparing to a scalar

- p4 eq 6: minus sign missing here two?

- p4 l8: maybe mention "outward pointing"

- p6 l27: "is the control parameter." (period missing)

- p7 l11: $\exp\left(\zeta\left(x,y\right)\right)$. (parenthesis missing)

- p8 l4: with respect to the initial input

- p8 eq 26 and 27: I think you should use capital $B_i$ at the numerator since your are deriving the function, not its output. Equation 26 should therefore be

$$\delta J = \left(\prod_{i=N}^{1} \frac{\partial B_i}{\partial b_{i-1}}\right) \frac{\partial B_0}{\partial \phi_i} \delta \phi_i \tag{6}$$

- p8 l27: to generate a *derivative*?

- p8 l29: derivative?

- p9 l2: gradient of *a* function

- p9 l6: forward accumulation AD tool: I think this method is generally referred to as "Object Overloading"

- p12 l9: 500 m (space missing)

- p13 l2: An L-curve analysis

- p15 figure 14: I think what matters is not so much that the sliding laws 2 and 3 are non linear, what is important here is that they depend on the effective pressure, so I would replace the third box to "Inversion: effective pressure-dependent sliding Law".

- p16 l2: 500 m (space missing)

- p17 l1: the the

- pp17 l15: maybe mention water sheet thickness?

- p17 l17: is it really mPa or MPa?

- p18 l10: Figure 14 and 17 (no parentheses needed)

- p22 l2: will account for some of the effects, which would (comma missing)

- p23 l1: hydrology runs are reflective

- p28 l6: we would like to thank M. ...
* * *

---

## Referee Comment (RC2) · Anonymous Referee #2 · 2 Oct 2017

General comments:

This study is an investigation of hydrologically-forced ice-flow model initialization using multiple inversions for basal drag. It explores three commonly-used sliding-law formulations in attempting to initialize seasonal runs with an end-of-winter hydro-mechanical state. The general scientific question addressed is worthwhile for all the usual reasons of improving model fidelity to observations and the need for practical and sensible means of incorporating the effects of basal hydrology on ice-sheet dynamics. The paper is clearly written.

The paper appears to report on part of a PhD thesis that seems a fulsome combination of model development, numerical implementation and glaciological application. Presumably for this reason, the paper has excessive detail in some places (particularly

where the model development appears to mimic previous work) and omission of detail elsewhere where it would be warranted. The paper could also make better use of space with many of the figures. A related consequence of the paper's origin is that it skates over the scientific justification for the development of a new ice-flow model that seems to implement what is already in the literature. One can imagine the reason for this: the author(s) coded this part of the model from scratch, but used existing code for the coupled hydrology. This is an excellent experience for a PhD student, but now the task of the authors is to justify to the scientific community why the world needs another ice-flow model, and this one in particular.

One of the main results of the paper is that using a Coulomb-friction-type sliding law, with a modelled distribution of effective pressure, yields a markedly different distribution of basal drag (and therefore sliding rate) than using a linear sliding law. This result is closely tied to the behavior of the hydrology model, and presumably to the parameters used in the sliding law. The differences are explained in terms of the non-linearity of the sliding law and its sensitivity to effective pressure, as well as the continuum nature of the hydrology model. The dependence of this result on model details warrants more emphasis on the parameters chosen for the hydrology model and Coulomb-friction sliding law, as well as the behavior of the latter.

This is a worthwhile study and I hope the comments below serve to improve the final paper.

Introduction of a new ice-flow model:

It appears that this depth-integrated model closely follows the work summarized in two sources (Goldberg, 2011; Arthern et al., 2013), with the novelty that the new model allows periodic boundary conditions (related to the ISMIP-HOM experiments). The authors even acknowledge that their model is more limited in some ways due to software (bottom of pg 6). Are there other departures from the two sources that could be highlighted as new innovations? How does this formulation differ or improve upon

the coupled (also depth-integrated, if I recall) model of Hewitt (2013), whose hydrology model is employed in this study?

For the problem presented in this paper (a single season and a single catchment), one might legitimately ask why it wouldn't be better to simply use an existing code like Elmer/Ice, which includes a built-in inversion for basal friction and may well also include the hydrology model of Werder et al (2013):

http://elmerice.elmerfem.org/wiki/lib/exe/fetch.php?media=courses:elmerice_2015_friction.pdf

Imbalanced detail:

The basic governing equations, simplifications, boundary conditions and sliding-law formulations given on pp 3-5 are needed, but section 2.1.2 (Implementation) could be condensed, as it seems to closely follow Arthern et al (2015). Section 2.2 (Inversion) is long and detailed, particularly considering that it seems to closely follow Goldberg and Heimbach (2013). For example, the information in the text on pg 7, lines 1-18, is pretty standard fare for inversions, so could be shortened. Section 2.2.2 is detailed and didactic; is the discussion of the TLM necessary? It is nice to have a brief description of the adjoint model, but I expected most readers would be somewhat familiar with these methods already.

On the other hand, the hydrology model is fundamental to this study but is only briefly described (p 10, lines 2-10). The hydrology model seems as important as the numerical details of the ice-sheet model. Consider presenting the key governing equations here. Although the equations are currently absent, the hydrology model includes parameters whose values must play an important role in the results (p. 13, lines 7-9). It would be worthwhile reporting values for the cavity step height, the effective hydraulic conductivity/permeability and the incipient channel-width length scale, along with any other parameter settings that differ from Hewitt (2013) and Banwell et al (2016). Further, the results and discussion would be more accessible if the reader knew a bit more about what went on with the hydrology model behind the scenes. For example, see p.

22, lines 4-6.

Space:

Consider moving the two blocks of pseudocode into an Appendix. Likewise, the flow chart in Figure 4 could be omitted.

There is a fair bit of blank space and redundancy in some of the figures. Here are some suggestions for a more efficient and impactful presentation:

- Combine Figs 2 and 3 (unlabeled E, N coordinate values can be removed from axis tick labels, as long as there is a scale bar) - Omit Figs 7a, 13a, 16a, or make them small insets in the corresponding b panels. - Omit or move to an appendix Fig 6. Nice to know how convergence occurs, but not necessary to show as a figure. - Combine Figs 8, 14, 17 into a single figure with 9 panels. This facilitates comparison. - Combine Figs 5, 12, 15. Could be done in a single panel figure. - Omit Fig 10. So much white space that could be replaced by a sentence. If it must be retained, consider a log plot.

Specific comments (page.line):

1.8: "a recent subglacial hydrology model" This sounds like it must be a different model than is used in the paper, but by the end it is clear that the model is that of Hewitt (2013). Please reword to clarify.

3.16 "Ab is the creep parameter set to an appropriate value for basal ice". One should explain why the flow-law rate factor should be different by an order of magnitude (see Table 1) for basal ice, particularly in light of the differences in the results between the Coulomb-friction sliding law (which uses Ab) and the other two sliding laws.

12.14: "the magnitude of the change is relatively limited" Pretty vague. Can this be quantified?

16.15: Why choose $\gamma_2 = 10^{-12}$ rather than $\gamma_2 = 10^{-11}$ in this type of trade-off curve?

17.15: "bed roughness scale of 0.5 m" refers to lambda_b?

21.Fig 11b: Consider plotting pw/(rho_ice g h_ice) rather than (or in addition to) N, as N does not immediately reveal how close the bed is to flotation.

21.8-9: It seems intuitive that there would be a contribution from deformation, so what is going wrong in the simulations/inversions to produce a better match of observed and modelled surface velocities when the sliding ratio (assuming that means Ub/Us) approaches one (i.e. plug flow)? Is it entirely explained by the assumption of uniform A? It seems that A_b would be playing a key role here, as mentioned in lines 6-7. A_b influences the sliding speed in the Coulomb friction law, but the value of the flow-law coefficient that regulates creep closure is probably A in the model formulation (or is it A_b?).

22.16-17: It would be compelling here if the authors could help the reader identify the effective pressures at which the behavioral transition in the sliding law occurs and relate them to the effective pressures shown in Fig 11b.

23. The conclusion that the modelled hydrology more resembles summer than winter conditions is not incorrect, but not especially meaningful. If the fragmentation of the drainage system is, in reality, what permits water storage in areas of high topography, then of course a continuum model fails to capture this effect. The authors acknowledge as much, but it diminishes the value of presenting this as a finding or conclusion of the study (as reported in the Abstract).

Technical corrections/queries (page.line):

1.13-14 "evolution of THE subglacial system" 1.16-17: "result IN faster flow" 2.10: AND missing 2.13: "one" should be "some" 5.1 "integrating" => "integrated" 5.12: Looks like u_b should be \bar{u} in Eqn (17) 8.4: "with respect to in the" too many words 14.3: "show" => "shown" 17.2: "the the" 19.Fig8c: Us/Ub? Sliding ratio sounds like it should be Ub/Us or Ub/Udef. 25.5 "model output of the model"

---

## Author Comment (AC1) · 16 Oct 2017

Reply to Reviewer 1

We thank the reviewer for their thoughtful evaluation of our paper, and the comments offered to improve it. Our replies follow. We use bold face for comments, normal face for our reply, and put changes to the manuscript in quotation marks, with italics indicating new text. Strikethrough text denotes deletion.

**This manuscript describes how to incorporate the effective pressure computed by a subglacial hydrology model into a basal drag inversion. Three different sliding laws are used: (1) a linear sliding law that does not depend on the effective pressure, (2) a "Budd"-type sliding law and a (3) a Schoof sliding law that both depend on the effective pressure. The authors propose here to do an inversion based on the first sliding law (1), to get a good initial guess of basal sliding, then run a subglacial hydrology model for the winter season, which computes the basal melt rate and basal water pressure that is then used to invert for the basal friction parameters used by the two other laws (2) and (3). They find that the basal drag and sliding ratio (ratio between the surface and basal velocities) of the linear and Budd sliding laws are in good agreement, but that their results using the Schoof sliding law show higher spatial variability.**

**Overall, this manuscript is easy to read and the methods are explained in detail, so that this manuscript is accessible to readers that are not necessarily familiar with model inversions. While I enjoyed reading it, I would have liked to see more discussion on their results instead of focusing mainly on the technical aspect. In its current state, the manuscript is limited to the introduction of a new model and a new approach, which would be more suitable to GMD for example. I do think that there is potential here for more scientific discussion. I am also a bit puzzled by their results with the third sliding law (see below), and why the slip ratios are so different. Since the same viscosity parameters are used for all three models, if the basal velocities are the same, the surface velocities should also be the same. Since we are trying here to reduce the misfit between InSAR velocities and modeled surface velocities, I would expect to see the same basal sliding velocities in all three cases, and this is not what is found here.**

**1 Major comments**

**I have one main concern, I don't understand the results of the Schoof sliding law. I read several times the explanations (page 22), but I still don't understand why the results are so different.**

**First, and maybe I am wrong, I don't think we need to do a second inversion once we have a good estimate of basal drag (τb) and basal velocities (ub) because there are always ways to change the friction parameters of different sliding laws to end up with the same basal velocities. If this is achieved, then the surface velocities are the same since the internal deformation is the same irrespective of the sliding law. In other words, one can invert for basal friction using a linear sliding law, and then use the results of a subglacial hydrology model to constrain the parameters of a different law using the results of the first inversion without performing another inversion.**

**For example, here, the three sliding laws are:**

$$\boldsymbol{\tau_b} = \beta^2 \boldsymbol{u_b}$$
$$\boldsymbol{\tau_b} = \mu_a N_+^p U_b{}^q \frac{\boldsymbol{u_b}}{U_b}$$
$$\boldsymbol{\tau_b} = \mu_b N_+ \left( \frac{U_b}{U_b + \lambda_b A_b N_+^n} \right)^{\frac{1}{n}} \frac{\boldsymbol{u_b}}{U_b}$$

**and the authors invert for β in (1), μa in (2) and μb in (3). If we invert for β and compute Ub and N+, one can determine μa and μb by simply doing:**

$$\mu_a = \frac{\beta^2 U_b}{N_+^p U_b^q}$$

**and**

$$\mu_b = \frac{\beta^2 U_b}{N_+ \left( \dfrac{U_b}{U_b + \lambda_b A_b N_+^{\frac{1}{n}}} \right)^n}$$

**Using these values for μa and μb, the forward model should produce the same surface velocities and therefore the same misfit to observations. The only problem might be the smoothness of these fields. So, I guess I have 2 questions:**

**Why do we need such a complicated procedure, when a simple single inversion would potentially enough?**

This was our initial approach. However, the issue of smoothness led us to consider whether the values determined using the algebraic approach reflected conditions at the ice sheet bed.

The aim of an inversion is not to simply find a field of basal drag that can reproduce surface velocities. Inversions problems are underdetermined, and there are an infinite number of possible solutions. Rather, the aim is to set-up the inversion problem such that the values of μa/μb are representative of physical properties of the bed. The constraint we have to solve this aim is that those properties need to be able to reproduce surface velocities. This is important as we run the model into the future, and use the inverted values in our coupled model to predict surface velocities through a summer melt season.

The linear inversion problem (as we formulate it, although different regularization schemes are possible) is to find the field $\beta^2$ that reproduces surface velocities, yet is sufficiently smooth. If we simplify the Budd sliding law by assuming p, q= 1, and then match the results to the linear sliding law, we are solving:

$$\beta^2 = \mu_a N \qquad (1)$$

We don't think this is a physical relationship. It is illustrative to look at the limit of N→0 (e.g. a subglacial lake). In that case, this formula tells us that μa→Inf. We don't interpret this value of μa as reflecting physical properties of the bed. A nearby cell where N ~ 1MPa would have a much different value of μa. However, geologically, we would anticipate the properties of the bed beneath a lake and at a nearby location to be similar. Similarly the results of the inversion should be resilient to small changes. If we invert before a subglacial lake drains, and after a subglacial lake drains, we would want to recover similar values of μa/μb in that location. This is not possible with the algebraic approach. By using Equation 1, the results also take on the assumptions made by the regularization, that the product of effective pressure and bed properties should vary smoothly. We feel that this isn't the best assumption we can make.

The approach we propose states a different problem. We assume we know N. We invert for μa/μb. We still have to regularize the problem, but in this case, the regularization is stating that the bed properties vary smoothly. This is a more physical and justified assumption in our perspective. In areas where N→0, the regularization enforces that μa/μb vary smoothly. It also possible that we may at some point have a-priori information about μa/μb or N via measurements. This approach would allow us to incorporate this information via different/additional regularization. For instance, rather than assume μa/μb vary smoothly, we may want to minimize the difference from a background value.

Each of the inversions for the different sliding laws is a different problem. Here, we are using the linear inversion to be able to predict N. Then we incorporate this information into the problem relating to the values of μa/μb. The algebraic approach may be a practical alternative for predicting the values of μa/μb under conditions when the magnitude of N has limited variability and varies smoothly. Although the effective pressures shown in the manuscript have those properties, the output of the subglacial hydrology model is sensitive to the parameters used. With plausible parameters, we observed N→0 and vary by orders of magnitude. This required us to generate a general approach. A continuum model of subglacial hydrology also leads to a smooth solution. Different formulations of subglacial hydrology may result in higher and more rapid variations of N.

On a practical note using Equation 1, with N=0, leads to μa=NaN. You have a problem of infilling these locations. This does not happen in our procedure, as in this case, μa/μb have no impact on the first term of the cost function (least squares misfit), and are determined by the second term in the cost function (smoothness).

**Why are the results of the Schoof sliding law so different? Is it because the inversion converged in a local minimum?**

Comments from both reviews led us to scrutinize our explanation in greater detail. Previously we had done a simple calculation which showed that N was in the order of magnitude where the sliding law was in the Coulomb limit. However, when responding to the comments, we plotted the transition point for a range of N and U, and found that the Schoof sliding law was not in the Coulomb limit for the range of velocities and effective pressures in our domain.

Unfortunately, there was a discrepancy between the text and figure. While wrote in the text that the results were from \gamma_2=1e11, we had plotted the results of \gamma_2=1e09. The higher frequencies observed are not due to the sliding law, but because of a discrepancy in regularization. We have updated the figure with the results we intended to show, which are now similar to the other sliding laws.

Text has been updated as follows:

[Non-linear sliding laws subsection of Results] Page 18, 10-11 Lines + Page 21 Lines 1-3

"~~Figure (14 and 17) show the inversion results from the Weertman and Schoof sliding law respectively. Inverted basal drag using the Weertman sliding law is very similar to the results from the linear sliding law. In contrast, the inverted basal drag from the Schoof sliding law shows much higher frequency and magnitude spatial variations. This is reflected in the spatial distribution of the sliding ratio, with the Weertman sliding law resulting in a distribution similar to the linear sliding law, while the distribution from the Schoof sliding law shows much greater variation.~~"

*"Figure 5 shows the inversion results from the Budd and Schoof sliding laws. Inverted basal drag and the consequent sliding ratio using the two non-linear sliding laws are very similar to the results from the linear sliding law. Model mismatch is also similar for all three sliding laws (Figure 4)."*

[Results Section] Page 22, Lines 7-19

"~~The pattern of basal drag inverted using the linear and Weertman sliding law show limited differences. This is due to the 10 fact that basal shear traction must satisfy the global stress balance (Joughin et al., 2004; Minchew et al., 2016). Both the linear and Weertman sliding law have the form τb = C · u1/m in the inversion, since effective pressure can be incorporated into the constant C for the Weertman sliding law. Previous work shows that in this case C ∝ u−1/m, and the recovered fields of basal drag are within a few percent of each other (Minchew et al., 2016). The basal drag and basal velocities from the the linear sliding law to initiate the subglacial hydrology model are therefore self consistent with the subsequent inversion results of 15 the Weertman sliding law. The pattern of basal drag inverted using the Schoof sliding law however, shows both higher spatial variability and a higher magnitude of variability. This is a result of the Schoof sliding law shifting to Coulumb-like behaviour at low effective pressures.~~"

*"The pattern of basal drag inverted using the three different sliding laws show limited differences. This is due to the fact that basal shear traction must satisfy the global stress balance \citep{Joughin2004, Minchew2016}. The basal drag and basal velocities from the linear sliding law used to initiate the subglacial hydrology model are therefore self consistent with the subsequent inversion results of the two non-linear sliding laws. For the winter effective pressures predicted by the subglacial hydrology model, we find that the Schoof sliding law is in the viscous drag regime. For a representative basal velocity of 75 \unit{m yr^{-1}}, the transition to Coulomb friction occurs at effective pressures of approximately 0.7 MPa. This is below modelled effective pressures, which are above 1.3MPa for most of the study domain."*

**It would be nice to try and start with the μa and μb from equations 4 and 5 and see if you indeed get the same sliding ratio for all 3 sliding laws.**

We hope that our discussion above illustrates that while this can be done on a technical level, the approach we propose is more general.

**2 Specific comments**

- **p3 equations 3, 4 and 5: I think you are missing a minus sign for all these equa- tions (basal drag opposes motion)**

  This form appears commonly in the literature [e.g Hewitt (2013)]. We subtract basal drag in the momentum equations.

- **p3 eq 5: use \left( \right) rather than simple parentheses.**

  Fixed

- **p3 l20: I would rather call this equation a Budd sliding law since he is the one**

  **who introduced effective pressure in basal stress.**

  Updated throughout.

- **p3 l24: you should take the norm of τb here (not the vector) since you are com- paring to a scalar**

  Fixed

- **p4 eq 6: minus sign missing here two?**

  See note above

- **p4 l8: maybe mention "outward pointing"**

  Included

- **p6 l27: "is the control parameter." (period missing)**

  Fixed

- **p7 l11: exp (ζ (x, y)). (parenthesis missing)**

  Fixed

- **p8 l4: with respect to the initial input**

  Updated

- **p8 eq 26 and 27: I think you should use capital Bi at the numerator since your are deriving the function, not its output. Equation 26 should therefore be**

$$\delta J = \left( \prod_{i=N}^{1} \frac{\partial B_i}{\partial b_{i-1}} \right) \frac{\partial B_0}{\partial \phi_i} \delta \phi_i$$

  updated equations

- **p8 l27: to generate a *derivative*?**

  Fixed

- **p8 l29: derivative?**

  Fixed

- **p9 l2: gradient of *a* function**

  Fixed

- **p9 l6: forward accumulation AD tool: I think this method is generally referred to**

  **as "Object Overloading"**

  Forward accumulation refers to the general idea/concept, while "Object Overloading" refers to a specific method of implementing forward accumulation.

- **p12 l9: 500 m (space missing)**

  Fixed

- **p13 l2: An L-curve analysis**

  Fixed

- **p15 figure 14: I think what matters is not so much that the sliding laws 2 and 3 are non linear, what is important here is that they depend on the effective pressure, so I would replace the third box to "Inversion: effective pressure-dependent sliding Law".**

  This is a good point. We have removed the flow chart in response to comments from Reviewer 2, but have incorporated this comment in a manuscript in preparation.

- **p16 l2: 500 m (space missing)**

  Fixed

- **p17 l1: the the**

  Fixed

- **pp17 l15: maybe mention water sheet thickness?**

  Updated to "The maximum distributed system sheet thickness"

- **p17 l17: is it really mPa or MPa?**

  Mpa, fixed

- **p18 l10: Figure 14 and 17 (no parentheses needed)**

  Fixed

- **p22 l2: will account for some of the effects, which would (comma missing)**

  Fixed

- **p23 l1: hydrology runs are reflective**

  Fixed

- **p28 l6: we would like to thank M. ...**

  Fixed

---

## Author Comment (AC2) · 16 Oct 2017

We are pleased that reviewer thought this was a worthwhile study, and appreciate the constructive comments. Our replies follow. We use bold face for comments, normal face for our reply, and put changes to the manuscript in quotation marks, with italics indicating new text. Strikethrough text denotes deletion.

**General comments:**

**This study is an investigation of hydrologically-forced ice-flow model initialization using multiple inversions for basal drag. It explores three commonly-used sliding-law formulations in attempting to initialize seasonal runs with an end-of-winter hydro-mechanical state. The general scientific question addressed is worthwhile for all the usual reasons of improving model fidelity to observations and the need for practical and sensible means of incorporating the effects of basal hydrology on ice-sheet dynamics. The paper is clearly written.**

**The paper appears to report on part of a PhD thesis that seems a fulsome combination of model development, numerical implementation and glaciological application. Presumably for this reason, the paper has excessive detail in some places (particularly where the model development appears to mimic previous work) and omission of detail elsewhere where it would be warranted. The paper could also make better use of space with many of the figures. A related consequence of the paper's origin is that it skates over the scientific justification for the development of a new ice-flow model that seems to implement what is already in the literature. One can imagine the reason for this: the author(s) coded this part of the model from scratch, but used existing code for the coupled hydrology. This is an excellent experience for a PhD student, but now the task of the authors is to justify to the scientific community why the world needs another ice-flow model, and this one in particular.**

**One of the main results of the paper is that using a Coulomb-friction-type sliding law, with a modelled distribution of effective pressure, yields a markedly different distribution of basal drag (and therefore sliding rate) than using a linear sliding law. This result is closely tied to the behavior of the hydrology model, and presumably to the parameters used in the sliding law. The differences are explained in terms of the non-linearity of the sliding law and its sensitivity to effective pressure, as well as the continuum nature of the hydrology model. The dependence of this result on model details warrants more emphasis on the parameters chosen for the hydrology model and Coulomb-friction sliding law, as well as the behavior of the latter.**

**This is a worthwhile study and I hope the comments below serve to improve the final paper.**

**Introduction of a new ice-flow model:**

**It appears that this depth-integrated model closely follows the work summarized in two sources (Goldberg, 2011; Arthern et al., 2013), with the novelty that the new model allows periodic boundary conditions (related to the ISMIP-HOM experiments). The authors even acknowledge that their model is more limited in some ways due to soft- ware (bottom of pg 6). Are there other departures from the two sources that could be highlighted as new innovations? How does this formulation differ or improve upon the coupled (also depth-integrated, if I recall) model of Hewitt (2013), whose hydrology model is employed in this study?**

**For the problem presented in this paper (a single season and a single catchment), one might legitimately ask why it wouldn't be better to simply use an existing code like Elmer/Ice, which includes a built-in inversion for basal friction and may well also include the hydrology model of Werder et al (2013):**

**http://elmerice.elmerfem.org/wiki/lib/exe/fetch.php?media=courses:elmerice_2015_friction.pdf**

The numerics of the ice sheet model are not particularly novel. The model differs from that of Hewitt (2013) in that the momentum equations are written in terms of depth integrated velocities rather than basal velocities.

There are a variety of reasons to write a new ice sheet model. The reviewer has identified a primary motivation, which is to understand the numerics/equations. For this study, the ice sheet model/inversion code needed to include sliding laws not implemented in existing models and be coupled with the subglacial hydrology model of Hewitt (2013). At the beginning of

the project, we were uncertain what this would entail in terms of links to any existing ice sheet model code, and pragmatically, writing the code from scratch ensured familiarity and flexibility of the code. Additionally, we have limited experience coding in low-level languages. Writing our own code allowed us to rapidly test model ideas/configurations, which could perhaps later be integrated in to more stable and full featured project such as ISSM / CISM / Elmer/ice. We haven't seen any publications from Elmer/Ice in regards to coupled modelling. Even so, subglacial hydrology model is in its early stage, and a diversity of models/approaches is beneficial to the community.

**Imbalanced detail:**

**The basic governing equations, simplifications, boundary conditions and sliding-law formulations given on pp 3-5 are needed, but section 2.1.2 (Implementation) could be condensed, as it seems to closely follow Arthern et al (2015). Section 2.2 (Inversion) is long and detailed, particularly considering that it seems to closely follow Goldberg and Heimbach (2013). For example, the information in the text on pg 7, lines 1-18, is pretty standard fare for inversions, so could be shortened. Section 2.2.2 is detailed and didactic; is the discussion of the TLM necessary? It is nice to have a brief description of the adjoint model, but I expected most readers would be somewhat familiar with these methods already.**

**On the other hand, the hydrology model is fundamental to this study but is only briefly described (p 10, lines 2-10). The hydrology model seems as important as the numer- ical details of the ice-sheet model. Consider presenting the key governing equations here. Although the equations are currently absent, the hydrology model includes parameters whose values must play an important role in the results (p. 13, lines 7-9). It would be worthwhile reporting values for the cavity step height, the effective hydraulic conductivity/permeability and the incipient channel-width length scale, along with any other parameter settings that differ from Hewitt (2013) and Banwell et al (2016).**

**Further, the results and discussion would be more accessible if the reader knew a bit more about what went on with the hydrology model behind the scenes. For example, see p. 22, lines 4-6.**

As the reviewer noted, there is a strong focus on the ice sheet model/inversion code details as this introduces a new code. However, the hydrology model is an 'off-the shelf' component.

We avoid detailed treatment of the hydrology model for a few reasons. The main innovation in this study from our perspective is the workflow, which is independent of the subglacial hydrology model. In this study, we use the model from Hewitt (2013) to generate the winter field of effective pressure. The parameters used are those we need to initiate summer runs based on calibration/validation of the integrated model using summer velocities. The winter results are an outcome of the summer calibration. Because the continuum model would not capture the fragmentation (as noted by the reviewer), the sheet thickness doesn't correspond to radar data, and due to a general lack of data on subglacial conditions, we don't scrutinize the winter hydrology output closely. Rather, the field of effective pressure is a-priori guess – plausible -- but not definitive for the above reasons.

This depends on your background, but our impression is that there is less familiarity with inversion procedures than subglacial hydrology in the community, as the latter lends itself to a conceptual understanding. From personal experience, understanding the methods behind inversions is involved. While the text was necessary for a thesis, our hope is that the in-depth writing could be useful for those without a background in adjoints. The crux in significantly cutting out sections would be that the paper remains verbose for experts, but not detailed enough for everyone else.

Overall, for reasons of length we think that only one of the models can be described in detail. There are good reasons for focusing on the ice sheet model/inversion code, although we acknowledge that some readers would benefit from the alternative choice suggested by this reviewer. Since reviewer one noted that our approach was accessible for readers without a background in inversions, we've decided to only make the minor changes suggested by reviewer one rather than rewrite the section.

**Consider moving the two blocks of pseudocode into an Appendix. Likewise, the flow chart in Figure 4 could be omitted. There is a fair bit of blank space and redundancy in some of the figures. Here are some suggestions for a more efficient and impactful presentation:**

Moved pseudocode to appendix, jettisoned flow chart.

**- Combine Figs 2 and 3 (unlabeled E, N coordinate values can be removed from axis tick labels, as long as there is a scale bar) - Omit Figs 7a, 13a, 16a, or make them small insets in the corresponding b panels. - Omit or move to an appendix Fig 6. Nice to know how convergence occurs, but not necessary to show as a figure. - Combine Figs 8, 14, 17 into a single figure with 9 panels. This facilitates comparison. - Combine Figs 5, 12, 15. Could be done in a single panel figure. - Omit Fig 10. So much white space that could be replaced by a sentence. If it must be retained, consider a log plot.**

We have:

A) Combined Figs 2 and 3.

B) Omitted figs 7a, 13a, 16a. Combined 7b, 13b, 16b into one figure with three panels.

Removed the text:

Page 17, Lines 1-6

"

Page 18, Lines 6-9

~~"The histogram of the absolute difference between observed and modelled surface velocities for both non-linear sliding laws shows a similar distribution to the linear sliding law inversion (Figure 13 and 16). The Weertman sliding law results in a spatial distribution of misfit similar to the linear sliding law, while spatial distribution of error from the Schoof sliding law shows higher frequency variations. Model mismatch again is highest in the vicinity of the nunatak.~~ "

C) Omitted the plot of convergence.

Removed the text:

Page 17, Line 1

"

D) Combined plots 8, 14, and 17

E) We have put figs 5,12, and 15 into a single plot with 3 panels. Overlaying them onto one panel obscured the scaling factors.

F) Omitted Figure 10

Removed the text:

Page 17, Line 13:

"

In addition, we have updated the colour palette in in some figures for consistency, removed the axis tick labels except in

Figure 1, and added a scale bar to figures which were lacking them.

**Specific comments (page.line):**

**1.8: "a recent subglacial hydrology model" This sounds like it must be a different model than is used in the paper, but by the end it is clear that the model is that of Hewitt (2013). Please reword to clarify.**

Apologies for the confusion, we hope this rewording of the last sentence eliminates the issue:

Additionally, we compare the modelled winter hydrological state to radar observations, and find that it is in line with summer rather than winter observations.

**3.16 "Ab is the creep parameter set to an appropriate value for basal ice". One should explain why the flow-law rate factor should be different by an order of magnitude (see Table 1) for basal ice, particularly in light of the differences in the results between the Coulomb-friction sliding law (which uses Ab) and the other two sliding laws.**

Amended as:

"Ab is the ice creep parameter *used for basal ice. It set an order of magnitude lower than A to account for warmer ice at the base (following Hewitt (2013)). The value of Ab is used in both the Schoof sliding law, and in the subglacial hydrology model for determining creep closure of channels and cavities. The value of A is used in the momentum equations.*"

**12.14: "the magnitude of the change is relatively limited" Pretty vague. Can this be quantified?**

The scale in the figure of Colgan et al. (2012) is quite coarse, while the data in Van de Wal et al. (2015) are averaged over several seasons. However, from the latter paper we make a conservative estimate of <25%. The text is changed as follows:

*"*the magnitude of the change is *limited (<25%)."*

**16.15: Why choose \gamma_2 = 10^{-12} rather than \gamma_2 = 10^{-11} in this type of trade-off curve?**

The L-curve is a method for selecting the trade-off between model fit and regularization. Although a common selection is the point of maximum curvature, this choice is subjective, and often results in over smoothing: (*https://www.sintef.no/globalassets/project/evitameeting/2005/lcurve.pdf*)

Our selection of \gamma_2 = 10^{-12} is subjective, although Morlighem (2013) selected a similar point on the L-curve.

Morlighem, M., et al. "Inversion of basal friction in Antarctica using exact and incomplete adjoints of a higher‐order model." *Journal of Geophysical Research: Earth Surface* 118.3 (2013): 1746-1753.

**17.15: "bed roughness scale of 0.5 m" refers to lambda_b?**

We were not clear with our terminology here. The 0.5 m refers to the cavity height scale used in the subglacial hydrology model, which controls the maximum sheet thickness. We chose omit this detail and to simplify this sentence so it reads:

"The maximum distributed system sheet thickness is 0.36 \unit{m}."

**21.Fig 11b: Consider plotting pw/(rho_ice g h_ice) rather than (or in addition to) N, as N does not immediately reveal how close the bed is to flotation.**

We think N is the appropriate field to plot as this is what we use in our sliding law. Since we don't discuss the hydrology model output in detail, plotting pw/(rho_ice g h_ice) isn't within the intended scope of the paper.

**21.8-9: It seems intuitive that there would be a contribution from deformation, so what is going wrong in the simulations/inversions to produce a better match of observed and modelled surface velocities when the sliding ratio (assuming that means Ub/Us) approaches one (i.e. plug flow)? Is it entirely explained by the assumption of uniform A? It seems that A_b would be playing a key role here, as mentioned in lines 6-7. A_b influences the sliding speed in the Coulomb friction law, but the value of the flow-law coefficient that regulates creep closure is probably A in the model formulation (or is it A_b?).**

Sliding ratio is Ub/Us. We fix figure 8C which had Us/Ub, and clarify this on page 14 with:
"and the sliding ratio ($\frac{U_b}{U_s}$) for the linear sliding law"

We use the value of A for the momentum equations in the ice flow model. We use A_b in the subglacial hydrology model, and the Schoof sliding law. (This has been clarified in the text in response to a previous question)

This issue is independent of the subglacial hydrology simulation, since the linear sliding law inversion does not depend on the hydrology. However, we're uncertain why stiffer ice leads to better fits in our inversions. We suspect that the topology of the minimization problem is more complex with a higher A (due to increasing possibility of tradeoff between basal sliding and internal deformation), leading to the inversion finding a local minimums.

**22.16-17: It would be compelling here if the authors could help the reader identify the effective pressures at which the behavioral transition in the sliding law occurs and relate them to the effective pressures shown in Fig 11b.**

Comments from both reviews led us to scrutinize our explanation in greater detail. Previously we had done a simple calculation which showed that N was in the order of magnitude where the sliding law was in the Coulomb limit. However, when responding to the comments, we plotted the transition point for a range of N and U, and found that the Schoof sliding law was not in the Coulomb limit for the range of velocities and effective pressures in our domain.

Unfortunately, there was a discrepancy between the text and figure. While wrote in the text that the results were from \gamma_2=1e11, we had plotted the results of \gamma_2=1e09. The higher frequencies observed are not due to the sliding law, but because of a discrepancy in regularization. We have updated the figure with the results we intended to show, which are now similar to the other sliding laws.

Text has been updated as follows:

[Non-linear sliding laws subsection of Results] Page 18, 10-11 Lines + Page 21 Lines 1-3

~~"Figure (14 and 17) show the inversion results from the Weertman and Schoof sliding law respectively. Inverted basal drag using the Weertman sliding law is very similar to the results from the linear sliding law. In contrast, the inverted basal drag from the Schoof sliding law shows much higher frequency and magnitude spatial variations. This is reflected in the spatial distribution of the sliding ratio, with the Weertman sliding law resulting in a distribution similar to the linear sliding law, while the distribution from the Schoof sliding law shows much greater variation."~~

*"Figure 5 shows the inversion results from the Budd and Schoof sliding laws. Inverted basal drag and the consequent sliding ratio using the two non-linear sliding laws are very similar to the results from the linear sliding law. Model mismatch is also similar for all three sliding laws (Figure 4)."*

[Results Section] Page 22, Lines 7-19

~~"The pattern of basal drag inverted using the linear and Weertman sliding law show limited differences. This is due to the 10 fact that basal shear traction must satisfy the global stress balance (Joughin et al., 2004; Minchew et al., 2016). Both the linear and Weertman sliding law have the form τb = C · u1/m in the inversion, since effective pressure can be incorporated into the constant C for the Weertman sliding law. Previous work shows that in this case C ∝ u−1/m, and the recovered fields of basal drag are within a few percent of each other (Minchew et al., 2016). The basal drag and basal velocities from the the~~

~~linear sliding law to initiate the subglacial hydrology model are therefore self consistent with the subsequent inversion results of 15 the Weertman sliding law. The pattern of basal drag inverted using the Schoof sliding law however, shows both higher spatial variability and a higher magnitude of variability. This is a result of the Schoof sliding law shifting to Coulumb-like behaviour at low effective pressures."~~

*"The pattern of basal drag inverted using the three different sliding laws show limited differences. This is due to the fact that basal shear traction must satisfy the global stress balance \citep{Joughin2004, Minchew2016}. The basal drag and basal velocities from the linear sliding law used to initiate the subglacial hydrology model are therefore self consistent with the subsequent inversion results of the two non-linear sliding laws. For the winter effective pressures predicted by the subglacial hydrology model, we find that the Schoof sliding law is in the viscous drag regime. For a representative basal velocity of 75 \unit{m yr^{-1}}, the transition to Coulomb friction occurs at effective pressures of approximately 0.7 MPa. This is below modelled effective pressures, which are above 1.3MPa for most of the study domain."*

**23. The conclusion that the modelled hydrology more resembles summer than winter conditions is not incorrect, but not especially meaningful. If the fragmentation of the drainage system is, in reality, what permits water storage in areas of high topography, then of course a continuum model fails to capture this effect. The authors acknowledge as much, but it diminishes the value of presenting this as a finding or conclusion of the study (as reported in the Abstract).**

We feel it doesn't detract from the abstract, and showing this empirically makes it worth reporting. It may be a less relevant result for those with a strong theoretical background.

**Technical corrections/queries (page.line):**

**1.13-14 "evolution of THE subglacial system"**

Fixed

**1.16-17: "result IN faster flow"**

Fixed

**2.10: AND missing**

Fixed

**2.13: "one" should be "some"**

Fixed

**5.1 "integrating" => "integrated"**

Fixed

**5.12: Looks like u_b should be \bar{u} in Eqn (17)**

Fixed

**8.4: "with respect to in the" too many words**

Removed 'in'

**14.3: "show" => "shown"**

Deleted.

**17.2: "the the"**

Deleted.

**19.Fig8c: Us/Ub? Sliding ratio sounds like it should be Ub/Us or Ub/Udef.**

Fixed

**25.5 "model output of the model"**

Fixed

---

## Author Response (AR2)

Dear Hilmar,

We are excited to see our manuscript accepted to The Cryosphere. We've uploaded an updated version, and reply to your minor comments below. Comments are in bold, our response is in normal text.

**I have a few minor comments that you might want to consider incorporating into the final version, but this I leave to you.**

**-I feel some of the citations give a bit distorted view of the literature. I would suggest citing the original work as much as possible. For example the SSA approximation is not as recent as 2011. Maybe cite the original paper my Morland or Physics of Glaciers instead?**

We have added references to Budd (1979), Schoof (2005) ,and Gagliardini et al. (2007) on Page 2, line 23 as references for the Budd and Schoof type sliding laws on Line 23. Removed the reference to Hewitt (2013) on Page 3, line 23

We have added the following italicized text at Page 2,line 30:

The ice sheet model implemented is based on the hybrid formulation described in Goldberg (2011) and Arthern et al. (2015),  and uses the numerical implementation of Arthern et al. (2015). *This formulation is derived from the Stokes Equations using variational principles (Dukowicz et al., 2010; Goldberg, 2011), and is a hybrid of the shallow ice approximation (Cuffey and Paterson, 2010) and the shallow shelf approximation (MacAyeal, 1989; Morland, 1987).*

**-Eq (11) seems to imply that \sigma_{xz}=0 for s=z which is not correct in the SSA approximation.**
This arises from the SIA part of the hybrid approximation.

**-You mention a priori knowledge, but I failed to see where that is introduced, or is the assumption \nabla \alpha_{a priori}=0 ? Generally one might expect a prior info to be introduced through a a-prior field and a corresponding covariance matrix as has been done in several Bayesian inversion studies in glaciology in the past.**

We have removed the following paragraph on page 7, line 7. It should avoid this confusion, and its probably unnecessary to describe weighted least squares.

[revised manuscript text omitted]